# Transport and Merge:
# Cross-Architecture Merging for Large Language Models

Chenhang Cui[1,*], Binyun Yang[2,*], Fei Shen[1,†], Yuxin Chen[1], Jingnan Zheng[1],
Xiang Wang[3], An Zhang[3], Tat-Seng Chua[1]

[1]National University of Singapore (NUS), Singapore
[2]University of Electronic Science and Technology of China (UESTC), China
[3]University of Science and Technology of China (USTC), China

## Abstract

Large language models (LLMs) achieve strong capabilities by scaling model capacity and training data, yet many real-world deployments rely on smaller models trained or adapted from low-resource data. This gap motivates the need for mechanisms to transfer knowledge from large, high-resource models to smaller, resource-constrained targets. While model merging provides an effective transfer mechanism, most existing approaches assume architecture-compatible models and therefore cannot directly transfer knowledge from large high-resource LLMs to heterogeneous low-resource targets. In this work, we propose a cross-architecture merging framework based on optimal transport (OT) that aligns activations to infer cross-neuron correspondences between heterogeneous models. The resulting transport matrices are then used to guide direct weight-space fusion, enabling effective high-resource to low-resource transfer using only a small set of inputs. Extensive experiments across low-resource languages and specialized domains demonstrate consistent improvements over target base models. The code is available at https://github.com/chenhangcuisg-code/Cross-Architecture-Merging-for-\Large-Language-Models/.

*Equal contribution. †Corresponding authors..

*Proceedings of the 43rd International Conference on Machine Learning*, Seoul, South Korea. PMLR 306, 2026. Copyright 2026 by the author(s).

## 1. Introduction

Large language models (LLMs) have achieved remarkable success in general language understanding and generation (Achiam et al., 2023; Grattafiori et al., 2024; Bai et al., 2023), enabling a wide range of downstream applications, e.g., in medicine (Xie et al., 2024), finance (Konstantinidis et al., 2024), and education (Wen et al., 2024). This success is largely driven by scaling model capacity and pretraining on massive, high-quality, and diverse corpora.

In practice, many real-world deployments rely on models with limited parameter budgets trained or adapted from low-resource data. Typical examples include low-resource languages such as Malaysian languages (Hew et al., 2025) and Cantonese dialects (Liu, 2022). Due to constraints on both model capacity and data availability, models trained from low-resource languages are unable to acquire specific knowledge comparable to large and data-rich models. A natural direction to address this challenge is high-resource to low-resource transfer (Adimulam et al., 2022; Myakala & Naayini, 2023; Cao et al., 2025), which leverages representations learned by large, well-trained models to improve performance in low-resource domains. Among existing approaches, model merging (Wortsman et al., 2022a; Jin et al., 2022; Imfeld et al., 2023a) has emerged as an alternative to gradient-based adaptation. It directly aggregates parameters from multiple expert models, enabling capability fusion without access to massive training data or repeated optimization. With the rise of large language models (LLMs) (Yang et al., 2024; Tao et al., 2024; Perin et al., 2024), this idea naturally extends to LLM merging.

Despite its appeal, many current techniques (Tao et al., 2024; Perin et al., 2024; Tian et al., 2025) assume that LLMs being merged share the same architecture, enabling direct weight-space operations. This assumption restricts their applicability in low-resource transfer scenarios, where the source model is often a large high-resource LLM, while the target model is a smaller architecture designed for deploy-

ment efficiency or domain-specific constraints (Liu et al., 2024; 2023a; Kolomeitsev, 2025). Direct parameter merging between heterogeneous architectures becomes challenging. Although some recent methods attempt cross-architecture merging or knowledge fusion for LLMs (Wan et al., 2024; 2025), they typically rely on distillation-based training (Hinton et al., 2015; Hsieh et al., 2023; Cai et al., 2025) to first distill knowledge into the parameter space of a target model, followed by merging among architecture-compatible models.

These limitations naturally raise the following question: **Can we enable high-resource to low-resource knowledge transfer across heterogeneous model architectures through direct model merging?** Motivated by this question, we propose a cross-model merging framework based on optimal transport (Peyré & Cuturi, 2019; Rout et al., 2021) to enable parameter transfer between heterogeneous architectures. Our approach is motivated by the observation that, despite differences in architecture and model capacity, language models often exhibit correlated internal activations when processing the same inputs (Huh et al., 2024; Shah & Khosla, 2025). Building on this insight, we establish a structured correspondence between heterogeneous models by aligning their internal activations using an optimal transport formulation. Given a small set of inputs, we extract intermediate activations from both the source and target models and compute cross-model similarity signals at the feature level. These signals are used to infer cross-neuron relationships between the two models, yielding a global transfer plan that captures transferable structure across layers and neurons. The resulting transport relationships are then converted into weight-space fusion operators, enabling direct parameter fusion across architectures.

In summary, our contribution is a cross-architecture model merging framework that enables high-resource to low-resource knowledge transfer. By aligning heterogeneous language models in activation space through an optimal transport formulation, our method infers structured neuron-level correspondences across models with different architectures and capacities using only a small set of inputs. Crucially, we show that these activation-space correspondences admit a principled weight-space interpretation, allowing them to be converted into direct fusion for parameter transfer across heterogeneous models. Extensive experiments on low-resource languages and domains demonstrate consistent improvements, establishing our approach as a practical alternative to distillation-based transfer. As shown in Figure 1, our method consistently improves performance across different settings under multiple strategies.

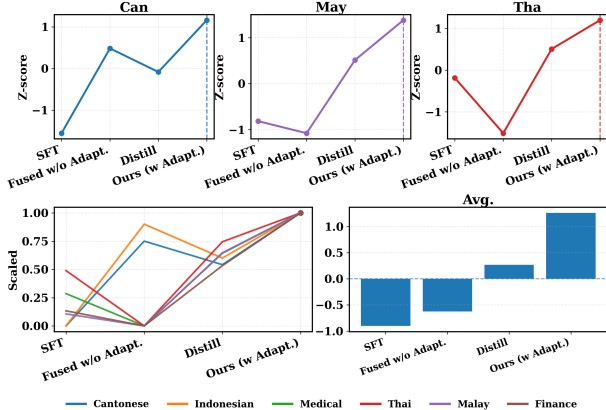

*Figure 1.* **Cross-domain comparison using normalized scores** (higher is better). **Top:** Domain-specific performance trajectories for Cantonese, Malaysian, and Thai under different transfer strategies. **Bottom-left:** Relative improvements within each domain. **Bottom-right:** Average normalized performance across all domains. See Section 5.3 for detailed analysis.

## 2. Related Work

**Model Merging.** Model merging seeks to combine multiple expert models into a single model, typically without access to the original training data and without costly training (Ilharco et al., 2022; Wortsman et al., 2022b; Ainsworth et al., 2022). Existing approaches primarily involve two stages: pre-merging and during-merging, covering both foundation models (LLMs and MLLMs) (Yang et al., 2024) and other deep learning settings (Li et al., 2023). Pre-merging methods focus on preparing models for subsequent fusion by modifying their parameters or representations in advance. Typical approaches include adaptation schemes designed to reduce parameter interference (Jacot et al., 2018; Ortiz-Jimenez et al., 2023) and alignment techniques that map different checkpoints into compatible parameter spaces prior to merging (Singh & Jaggi, 2020; Imfeld et al., 2023b). During-merging methods perform the actual fusion, ranging from simple parameter averaging (Wortsman et al., 2022a) to more structured strategies such as subspace-based merging (Yadav et al., 2023), as well as optimization-based merging (Wei et al., 2025). Although some methods attempt cross-architecture merging or knowledge fusion in LLMs (Wan et al., 2024; 2025), they generally rely on distillation-based training to first transfer knowledge into the parameter space of a target model.

**Optimal Transport and Applications.** Optimal transport (OT) provides a principled framework for comparing probability measures by finding a minimum-cost plan that transports mass from a source distribution to a target distribution (Peyré & Cuturi, 2019). Building on solid theory, OT has become a versatile tool across a broad range of applications. In machine learning, OT is commonly used for distribution matching in generative modeling (Rout et al., 2021), domain adaptation under distribution shift (Liu et al., 2023b; Redko et al., 2019), imitation learning (Luo et al.,

2023), and clustering (Del Barrio et al., 2019; Lin et al., 2023). More recently, OT has been explored as a mechanism for model merging, where a transport plan provides a soft correspondence between neurons and enables weight merging across models (Imfeld et al., 2023a; Singh & Jaggi, 2020). However, while recent studies have used optimal transport (OT) to reveal cross-model feature similarity even between heterogeneous LLMs (Shah & Khosla, 2025), how to effectively leverage OT for model merging with different architectures remains underexplored.

## 3. Preliminary

### 3.1. Optimal Transport

Optimal transport (OT) provides a principled way to compute a soft correspondence between two sets of objects under a pairwise cost. Given a cost matrix $C \in \mathbb{R}^{n \times m}$ and two discrete distributions $a \in \Delta_n$, $b \in \Delta_m$, OT solves

$$\min_{Q \in \mathbb{R}_+^{n \times m}} \langle C, Q \rangle \quad \text{s.t.} \quad Q\mathbf{1} = a, \; Q^\top \mathbf{1} = b. \quad (1)$$

We adopt the entropically regularized variant,

$$\min_{Q \in \mathbb{R}_+^{n \times m}} \langle C, Q \rangle - \varepsilon H(Q) \quad \text{s.t.} \quad Q\mathbf{1} = a, \; Q^\top \mathbf{1} = b, \quad (2)$$

where $H(Q) = -\sum_{i,j} Q_{ij}(\log Q_{ij} - 1)$ and $\varepsilon > 0$. The solution admits the Sinkhorn scaling form (Cuturi, 2013) $Q = \text{diag}(u) \, K \, \text{diag}(v)$ with $K = \exp(-C/\varepsilon)$, and can be computed by alternating updates on $u$ and $v$.

### 3.2. Neurons and Features in Transformers

In this paper, we adopt a structural view of neurons in Transformer models. Specifically, we use the term "neuron" to refer to a unit in the weight space of a linear sublayer, corresponding to a single row or column of the associated weight matrix. The values propagated through the network are referred to as activated features.

Specifically, consider a linear transformation

$$y = Wx, \quad (3)$$

where $W \in \mathbb{R}^{d_{\text{out}} \times d_{\text{in}}}$. We distinguish two neuron spaces induced by $W$: (i) input-side neurons correspond to the *columns* of $W$ (indexed by $d_{\text{in}}$), and (ii) output-side neurons correspond to the *rows* of $W$ (indexed by $d_{\text{out}}$). Accordingly, $x \in \mathbb{R}^{d_{\text{in}}}$ and $y \in \mathbb{R}^{d_{\text{out}}}$ represent features in the input and output spaces, respectively.

In a Transformer block, the primary linear sublayers include the attention projections

$$Q = W_Q h, \quad K = W_K h, \quad V = W_V h, \quad (4)$$

and the MLP projections

$$z = W_1 h, \quad h' = \sigma(z), \quad o = W_2 h', \quad (5)$$

where $\sigma(\cdot)$ denotes an elementwise nonlinearity. In all these cases, rows and columns of the projection matrices define neurons in weight space, while each dimension of the activations corresponds to a feature induced by these neurons.

## 4. Method

**Problem Definition.** Our goal is to merge models across heterogeneous architectures. We consider a target model $M_A$ with $L$ layers and a source model $M_B$ with $M$ layers.

Given a set of $T$ samples $\mathcal{D} = \{x_1, \ldots, x_T\}$, we record activations from both models across transformer modules:

$$\begin{aligned}
X_\ell &\in \mathbb{R}^{T \times n_\ell}, \quad \ell = 1, \ldots, L, \\
Y_m &\in \mathbb{R}^{T \times n'_m}, \quad m = 1, \ldots, M,
\end{aligned} \quad (6)$$

where each row corresponds to one input sample and each column corresponds to one feature channel. Details of the data used for activation extraction and transport estimation are provided in Appendix B.1.

### 4.1. Finding Feature and Layer Relationships using Optimal Transport

We use optimal transport (OT) (Peyré & Cuturi, 2019) to infer transport relationship matrices from activations, which reveal how activated feature channels in a source model can be recombined to match those in a target model. Although the ultimate goal is to merge parameters (i.e., neuron weights), directly establishing neuron-to-neuron correspondences is challenging across heterogeneous architectures. Instead, we first infer correspondences in the activation space and then lift these activation-level relationships to the neuron level for parameter transport.

Concretely, for each target layer $\ell$ and source layer $m$, we quantify cross-model similarity at the level of activation channels by constructing a correlation-based cost matrix

$$C^{\ell m}[i,j] = d_{\text{corr}}(X_\ell[:,i], Y_m[:,j]) \triangleq 1 - \rho(X_\ell[:,i], Y_m[:,j]), \quad (7)$$

where $\rho(\cdot, \cdot)$ denotes the Pearson correlation coefficient.

Based on this cost matrix, we compute a feature relationship matrix $Q^{\ell m} \in \mathbb{R}^{n_\ell \times n'_m}$ by solving an entropically regularized optimal transport problem:

$$Q^{\ell m} = \arg \min_{Q \in \mathcal{T}(n_\ell, n'_m)} \langle C^{\ell m}, Q \rangle - \varepsilon H(Q), \quad (8)$$

where $H(Q) = -\sum_{i,j} Q_{ij}(\log Q_{ij} - 1)$ and $\varepsilon > 0$. The transportation polytope enforces balanced marginals,

$$\mathcal{T}(n_\ell, n'_m) = \left\{ Q \in \mathbb{R}_+^{n_\ell \times n'_m} : Q\mathbf{1} = \tfrac{1}{n_\ell}\mathbf{1}, \; Q^\top \mathbf{1} = \tfrac{1}{n'_m}\mathbf{1} \right\}. \quad (9)$$

The resulting transport plan $Q^{\ell m}$ specifies how each target activation channel can be represented as a weighted combination of source channels. Although inferred purely from

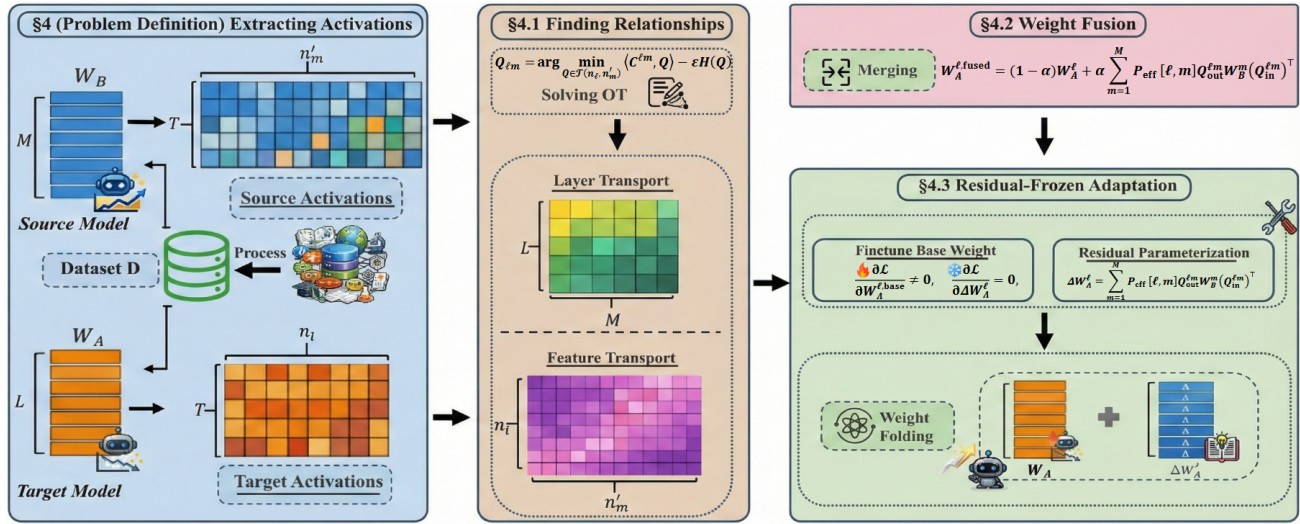

*Figure 2.* **Illustration of cross-architecture merging pipeline.** Given a small dataset $\mathcal{D}$, we extract intermediate activations from a high-resource source model and a low-resource target model with heterogeneous architectures. We then use optimal transport to infer layer- and feature-level correspondences, and leverage the resulting transport plans for direct parameter fusion. Finally, the fused model can be optionally refined via residual-frozen adaptation, where the transferred residuals are kept fixed and only base weights are updated.

activation statistics, $Q^{\ell m}$ will later be used as a neuron-level mixing operator for transporting source parameters into the target neuron coordinates.

To summarize the compatibility between target layer $\ell$ and source layer $m$, we aggregate the feature-level correspondence into a scalar transport cost inspired by (Shah & Khosla, 2025)

$$C_{\text{layer}}[\ell, m] = \langle C^{\ell m}, Q^{\ell m} \rangle, \qquad (10)$$

which yields a layer-to-layer cost matrix $C_{\text{layer}} \in \mathbb{R}^{L \times M}$.

Using these layer-wise costs, we further infer a global correspondence matrix $P \in \mathbb{R}^{L \times M}$ by solving another entropically regularized optimal transport problem:

$$P = \arg \min_{P \in \mathcal{T}(L,M)} \langle C_{\text{layer}}, P \rangle - \eta H(P), \qquad (11)$$

where $\eta > 0$ and

$$\mathcal{T}(L, M) = \left\{ P \in \mathbb{R}_+^{L \times M} : P\mathbf{1} = \tfrac{1}{L}\mathbf{1}, \ P^\top \mathbf{1} = \tfrac{1}{M}\mathbf{1} \right\}. \qquad (12)$$

Each entry $P_{\ell m}$ quantifies the degree of correspondence between target layer $\ell$ and source layer $m$. Together, the feature-level transport plans $\{Q^{\ell m}\}$ and the global matrix $P$ specify how source neurons are used to construct target neurons. All optimal transport problems are solved using Sinkhorn iterations (Cuturi, 2013); implementation details about the algorithm are provided in Appendix A.

## 4.2. From Activated Features to Neurons: Weight Fusion via Feature and Layer Transport

Our transport matrices are estimated from activated features, while the ultimate objective is to merge neurons and their associated parameters. The key observation is that each feature dimension corresponds to a well-defined neuron axis in the underlying weight matrix, specifically, a column in the input space or a row in the output space. This correspondence allows us to lift feature-level relations into neuron-level mixing operators, which are then used to transport source weights into the target neuron basis.

**Feature- and Layer-wise Transport.** In attention modules, input-side (pre-projection) and output-side (post-projection) features lie in different representation spaces and may induce different cross-layer alignment signals. Accordingly, we estimate two feature-level transport plans, $Q_{\text{in}}^{\ell m}$ from pre-activation features and $Q_{\text{out}}^{\ell m}$ from post-activation features. At the layer level, we compute two compatibility costs by aggregating the corresponding feature-level transport objectives, and solve the layer-level OT *separately* for the pre- and post-sides, yielding $P_{\text{pre}}$ and $P_{\text{post}}$, respectively. We then combine them into an effective layer correspondence

$$P_{\text{eff}}[\ell, m] = \sqrt{P_{\text{pre}}[\ell, m] \cdot P_{\text{post}}[\ell, m]}, \qquad (13)$$

**Selective transport via top-$k$ neuron replacement.** To improve robustness across heterogeneous architectures and avoid unnecessary interference, we restrict fusion to a small subset of neurons that are highly activated on the same dataset $\mathcal{D}$ used for alignment. Concretely, for each layer $\ell$ and each projection module, we run the target model on

$\mathcal{D}$ and record neuron activations via forward hooks. For each neuron, we compute an activation strength score as the mean absolute activation over samples $s_j = \frac{1}{T} \sum_{t=1}^{T} |h_{t,j}|$, where $h_{t,j}$ denotes the activation of neuron $j$ at sample $t$. We then select the top-$k$ neurons with the largest scores in each layer and projection module, and record their indices. Details about selection can be found in Appendix A.5

Fusion is applied only to these selected neuron indices, while all remaining parameters in the target model are left unchanged. For attention projections, this corresponds to replacing the rows or columns of the weight matrices associated with the selected neurons. This design yields a partial weight fusion scheme that injects source knowledge only into neurons that are both highly active on the target data and well-aligned across models. Under this design, the fused weights at target layer $\ell$ can be written as:

$$
\begin{aligned}
W_A^{\ell,\text{fused}} &= W_A^{\ell} \\
&+ \alpha \cdot \mathbf{M}^{\ell} \odot \left( \sum_{m=1}^{M} P_{\text{eff}}[\ell, m] \, Q_{\text{out}}^{\ell m} \, W_B^{m} \, (Q_{\text{in}}^{\ell m})^{\top} - W_A^{\ell} \right),
\end{aligned}
\tag{14}
$$

where $\mathbf{M}^{\ell}$ is a binary neuron-level mask that is nonzero only on the selected top-$k$ neuron indices, $\odot$ denotes masking at the neuron level, and $\alpha \in [0, 1]$ controls the strength of replacement. Bias terms are handled analogously.

### 4.3. Residual-Frozen Adaptation after Fusion

After fusion, only a small subset of neurons in the target model are modified by transported source parameters, while the remaining parameters remain identical to the original target model. During this stage we freeze the transported residual parameters and train only the original base parameters of the target model. This design ensures that the transferred knowledge is not overwritten by subsequent optimization, while still allowing the target model to adapt to the downstream task distribution. Empirically, this residual-frozen adaptation consistently outperforms training the target model alone under the same conditions, indicating that fusion provides a strong and stable initialization.

**Residual Parameterization.** We parameterize fusion through a residual form that separates transferred neuron-level knowledge from the original model parameters. Specifically, weight of target layer $\ell$ is expressed as

$$
W_A^{\ell,\text{fused}} = W_A^{\ell,\text{base}} + \alpha \cdot \mathbf{M}^{\ell} \odot \Delta W_A^{\ell}, \tag{15}
$$

where $W_A^{\ell,\text{base}}$ denotes the original weights of the target model $M_A$, $\mathbf{M}^{\ell}$ is a neuron-level mask that is nonzero only on the selected top-$k$ neurons, $\odot$ denotes neuron-level masking, and $\alpha \in [0, 1]$ controls the strength of replacement. The transported residual $\Delta W_A^{\ell}$ aggregates source operators

across layers via activation-aligned transport,

$$
\Delta W_A^{\ell} = \sum_{m=1}^{M} P_{\text{eff}}[\ell, m] \, Q_{\text{out}}^{\ell m} \, W_B^{m} \, (Q_{\text{in}}^{\ell m})^{\top}, \tag{16}
$$

where $P_{\text{eff}}[\ell, m]$ is the effective layer correspondence and $Q_{\text{in}}^{\ell m}, Q_{\text{out}}^{\ell m}$ are the feature-level transport maps inferred from activation statistics. Only the neurons selected by $\mathbf{M}^{\ell}$ participate in fusion; all others remain unchanged.

**Residual-Freezing and Adaptation.** During post-fusion adaptation, we freeze the transported residual $\Delta W_A^{\ell}$ entirely and optimize only the base parameters $W_A^{\ell,\text{base}}$. Gradients are therefore applied uniformly to the base weights, without any neuron-level freezing:

$$
\frac{\partial \mathcal{L}}{\partial \Delta W_A^{\ell}} = 0, \qquad \frac{\partial \mathcal{L}}{\partial W_A^{\ell,\text{base}}} \neq 0. \tag{17}
$$

This separation preserves the transferred neuron-level representations while allowing the target model to adapt globally.

**Weight Folding.** After adaptation, we fold the residual back into the base weights to obtain a parameter representation for inference:

$$
W_A^{\ell,\text{final}} = W_A^{\ell,\text{base}} + \alpha \cdot \mathbf{M}^{\ell} \odot \Delta W_A^{\ell}. \tag{18}
$$

This folding operation yields a model that is architecturally identical to the target model, with transferred neuron-level knowledge permanently absorbed into its parameters.

We summarize the overall transport-and-merge procedure with masked neuron replacement in Alg. 1.

### 4.4. Weight Transport as Representation-Space Transfer

Transporting source weights into the target model admits an exact interpretation in representation space: it is equivalent to mapping target features into the source feature coordinates, applying the source linear map there, and mapping the result back to the target space.

**Setup.** Fix a target sublayer $\ell$ in $M_A$ and a source sublayer $m$ in $M_B$ with linear operators (bias omitted)

$$
W_A^{\ell} \in \mathbb{R}^{d_{A,\text{out}}^{\ell} \times d_{A,\text{in}}^{\ell}}, \qquad W_B^{m} \in \mathbb{R}^{d_{B,\text{out}}^{m} \times d_{B,\text{in}}^{m}}.
$$

Let $Q_{\text{in}}^{\ell m}$ and $Q_{\text{out}}^{\ell m}$ be transport relationship matrices estimated from pre- and post-activation features. Define the induced coordinate maps

$$
\Phi_{\text{in}}^{\ell m} \triangleq (Q_{\text{in}}^{\ell m})^{\top}, \qquad \Phi_{\text{out}}^{\ell m} \triangleq Q_{\text{out}}^{\ell m}. \tag{19}
$$

We then define the transported source operator acting on target features as

$$
\widetilde{W}_{B \to A}^{\ell m} \triangleq \Phi_{\text{out}}^{\ell m} \, W_B^{m} \, \Phi_{\text{in}}^{\ell m}. \tag{20}
$$

**Algorithm 1** Transport and Merge for Cross-Architecture Model Merging

**Require:** Target model $M_A$ with $L$ layers, source model $M_B$ with $M$ layers, samples $\mathcal{D} = \{x_t\}_{t=1}^T$, fusion coefficient $\alpha$

**Ensure:** Fused target model $M_A^{\text{fused}}$

1: **Activation extraction:**
2: Run $M_A$ and $M_B$ on $\mathcal{D}$ to obtain activations $\{X_\ell\}_{\ell=1}^L$ and $\{Y_m\}_{m=1}^M$
3: **Feature-level transport:**
4: **for** each target layer $\ell$ and source layer $m$ **do**
5:     Construct cost matrix $C^{\ell m}$ (Eq. 7)
6:     Solve entropic OT to obtain $Q_{\text{in}}^{\ell m}, Q_{\text{out}}^{\ell m}$ (Eq. 8)
7: **end for**
8: **Layer-level transport:**
9: Compute pre/post layer costs from the corresponding $Q_{\text{in}}^{\ell m}$ and $Q_{\text{out}}^{\ell m}$ (Eq. 10)
10: Solve OT to obtain $P_{\text{pre}}$ and $P_{\text{post}}$ (Eq. 11)
11: Compute $P_{\text{eff}}$ (Eq. 13)
12: **Weight fusion:**
13: **for** each target layer $\ell$ **do**
14:     Fuse weights using Eq. 14
15: **end for**
16: **Optional post-fusion adaptation:**
17: Freeze transferred residuals, fine-tune base parameters, and fold residuals for inference (Eqs. 15–18)
18: **Output:** $\mathcal{M}_A^{\text{fused}}$

---

**Theorem 4.1** (Representation-Space Interpretation of Weight Transport). *For any target representation $h_A \in \mathbb{R}^{d_{A,\text{in}}^\ell}$ and any $(\ell, m)$,*

$$\widetilde{W}_{B \to A}^{\ell m} h_A = \Phi_{\text{out}}^{\ell m}\left(W_B^m\left(\Phi_{\text{in}}^{\ell m} h_A\right)\right), \quad (21)$$

Our fused target operator aggregates transported source operators across layers:

$$W_A^{\ell,\text{fused}} = (1 - \alpha)\, W_A^\ell + \alpha \sum_{m=1}^M P_{\text{eff}}[\ell, m]\, \widetilde{W}_{B \to A}^{\ell m}. \quad (22)$$

Combining Eq. 22 with Theorem 4.1 shows that fusion realizes a mixture of source-space computations driven by target features transferred into the corresponding source coordinate systems.

**Corollary 4.2** (Uniqueness Under Invertible Coordinate Maps). *If $\Phi_{\text{in}}^{\ell m}$ and $\Phi_{\text{out}}^{\ell m}$ are invertible, then $\widetilde{W}_{B \to A}^{\ell m}$ is the unique operator on the target space that reproduces the source-layer computation under the target-to-source coordinate transfer in Eq. 21.*

Proofs are provided in Appendix A.4.

---

*Table 1.* Low-resource language transfer results on MalayMMLU. Accuracy (%; higher is better).

| Category | base model | Fused w/o adaptation | Fused w/ adaptation |
|---|---|---|---|
| Humanities | 42.30 | 46.19 | **48.81** |
| Language | 38.37 | 40.52 | **41.68** |
| Others | 44.50 | 46.63 | **48.69** |
| STEM | 42.61 | 45.19 | **46.58** |
| Social Science | 40.92 | 43.77 | **46.91** |

*Table 2.* Low-resource language transfer results on Indonesian benchmarks. Accuracy (%; higher is better).

| Benchmark | Base Model | Fused w/o Adaptation | Fused w/ Adaptation |
|---|---|---|---|
| ARC (Indonesian) | 23.4 | **24.0** | 23.7 |
| Belebele (Indonesian) | 34.8 | 36.4 | **36.9** |
| TruthfulQA-MC (Indonesian) | 35.9 | 36.5 | **36.6** |
| XCOPA (Indonesian) | 59.6 | 59.8 | **60.0** |
| **Average** | 38.43 | 39.18 | **39.30** |

## 5. Experiments

### 5.1. High- to Low-Resource Language Transfer

**Setup.** We refer to the original unfused target model as the **Base Model**. We consider two variants of our method: Fused w/o Adaptation, which performs transport-guided parameter fusion without any post-training, and Fused w/ Adaptation, which applies an adaptation stage. To ensure experimental consistency, we fix the source architecture to the LLaMA-3 8B family (Grattafiori et al., 2024) in this section. Comparisons with other model families are deferred to later experiments in Section 5.4. We adopt four low-resource languages. **Malaysian:** Target is Malaysian-LLaMA-3-1B-Instruct (Mesolitica, 2025). **Indonesian:** Target is LLaMA-3-1B-Indonesian-QLoRA (Wahyurejeki, 2025). **Thai:** Target is Typhoon2-1B-Instruct (Pipatanakul et al., 2024). **Cantonese:** To better match linguistic proximity, we use a Chinese instruction-tuned source LLaMA3-Chinese-8B-Instruct (FlagAlpha, 2025), and fuse into the target model LLaMA-3-1B-Instruct. Details of the data used for adaptation and evaluation benchmarks are provided in Appendix B.1 and Appendix B.2, respectively.

**Malaysian.** Table 1 shows results of our method on MalayMMLU (Poh et al., 2024). We can observe that our method consistently achieves the best performance. Relative to the Base Model, Fused w/ Adaptation gains +6.51 (Humanities), +3.31 (Language), +4.19 (Others), +3.97 (STEM), and +5.99 (Social Science), indicating successful transfer from the high-resource language.

**Indonesian.** Table 2 reports results on Indonesian benchmarks (Clark et al., 2018; Bandarkar et al., 2024; Lin et al., 2022; Ponti et al., 2020). Fusion alone already improves most tasks, suggesting effective language capability transfer without extra training. With post-fusion adaptation, the model attains the best scores on three of four benchmarks, while Fused w/o Adaptation remains best on ARC.

*Table 3.* Low-resource language transfer results on CMMLU (Cantonese) (Jiang et al., 2025). Accuracy (%; higher is better).

| Category | Base Model | Fused w/o Adaptation | Fused w/ Adaptation |
|---|---|---|---|
| Humanities | 25.44 | **27.72** | 27.34 |
| Social Science | 25.07 | 27.10 | **27.36** |
| STEM | 25.22 | **26.63** | 26.63 |
| Others | 25.84 | **29.21** | 28.99 |
| **Average** | 25.26 | 27.41 | **27.44** |

*Table 4.* Low-resource language transfer results on Thai benchmarks. Score (higher is better).

| Benchmark | Base Model | Fused w/o Adaptation | Fused w/ Adaptation |
|---|---|---|---|
| MMLU (Thai) | 0.15 | 0.13 | **0.17** |
| MGSM (Thai) | 0.50 | 0.64 | **0.72** |
| XCOPA (Thai) | 0.58 | 0.58 | **0.60** |
| **Average** | 0.41 | 0.45 | **0.50** |

*Table 5.* General-domain to finance-domain transfer results on financial benchmarks. Score (higher is better).

| Benchmark | Base Model | Fused w/o Adaptation | Fused w/ Adaptation |
|---|---|---|---|
| MMLU (Business Ethics) | 0.36 | 0.37 | **0.38** |
| MMLU (Microeconomics) | 0.34 | **0.39** | 0.37 |
| MMLU (Professional Accounting) | 0.26 | 0.27 | **0.29** |
| **Average** | 0.32 | 0.34 | **0.35** |

*Table 6.* General-domain to medical-domain transfer results on medical benchmarks. Accuracy (%; higher is better).

| Benchmark | Base Model | Fused w/o Adaptation | Fused w/ Adaptation |
|---|---|---|---|
| MMLU (Anatomy) | 49.6 | 50.0 | **51.1** |
| MMLU (Medical Genetics) | 50.0 | 49.0 | **52.0** |
| MMLU (Professional Medicine) | 39.3 | 39.7 | **40.5** |
| **Average** | 46.30 | 46.23 | **47.87** |

stable consolidation of transferred medical knowledge.

### 5.3. Effectiveness of Cross-Architecture Merging

**Evidence of cross-architecture Representation Similarity.** We analyze representation similarity across heterogeneous models using the feature-level optimal transport plans $Q^{\ell m}$ learned during fusion (Eq. (8)). Each entry $Q_{ij}^{\ell m}$ measures the amount of activation mass transported from a source feature channel to a target feature channel, under balanced marginal constraints. To quantify how concentrated these correspondences are, we compute the percentage of transport mass captured by the top-$k$ entries of $Q^{\ell m}$. Specifically, for each $Q^{\ell m}$, we sort all entries in descending order and accumulate the transport mass of the top-$k$ entries, reporting this quantity as a percentage of the total transport mass. We then average this quantity across all layer pairs and modules. Figure 3 plots the averaged transport mass. The horizontal axis corresponds to the top-$k$ neuron correspondences, while the vertical axis shows the average percentage of source-to-target transport mass. Across all domains, a small number of neuron pairs explains a large fraction of the transport mass, indicating sparse yet aligned internal representations across heterogeneous LLMs.

**Performance Comparison with Merging Source Models of Different Sizes.** We further analyze how the capacity of the source model affects cross-architecture fusion performance. Using MalayMMLU as a representative benchmark, we merge Malaysian-LLaMA-3-1B-Instruct with general-purpose LLaMA models of different scales (1B, 8B, and 32B). Table 7 shows that increasing the source model size leads to consistent performance improvements. While merging with an 8B source already yields substantial gains across all categories, scaling the source to 32B provides additional improvements. This indicates that larger source models offer richer transferable representations that can be selectively injected through our neuron-level fusion mechanism.

**Compared with SFT Training and Distillation.** As illustrated in Figure 1, our method consistently outperforms both

**Cantonese.** Table 3 shows accuracy of CMMLU benchmark (Jiang et al., 2025). Fused w/ Adaptation improves all categories and raises the overall average from 25.26 to 27.44 (+2.18). The consistent gains suggest our fusion can transfer relevant knowledge despite different architectures.

**Thai.** Table 4 summarizes Thai benchmark (Hendrycks et al., 2020; Ponti et al., 2020; Shi et al., 2022) results. Fusion substantially boosts the MGSM task even without adaptation (0.50 → 0.64), and adaptation further improves all reported benchmarks, achieving the best overall scores. This indicates fusion provides effective transfer, while adaptation stabilizes and calibrates task performance.

### 5.2. General-to-Expert Domain Transfer

**Setup.** We also evaluate general-domain to expert-domain transfer. For **Finance**, we fuse LLaMA-3-1B-TEL-A-finance (TEL-LLM, 2025) with Qwen2.5-7B (Bai et al., 2023). For **Medical**, we fuse medical version LLaMA (Grattafiori et al., 2024) with LLaMA-3.2-8B.

**Finance.** Table 5 reports results on three finance-related MMLU subsets (Hendrycks et al., 2020). Direct fusion without adaptation largely preserves the Base Model's performance, yielding either marginal gains or near-parity across tasks, which indicates that cross-domain fusion does not introduce negative interference. With post-fusion adaptation, the fused model consistently achieves the best performance on all three benchmarks. These results suggest that fusion serves as an effective initialization that transfers general knowledge into the target domain.

**Medical.** Table 6 shows consistent improvements from fusion on medical benchmarks (Hendrycks et al., 2020). Even without adaptation, fusion yields positive gains on anatomy and professional medicine. Moreover, Fused w/ Adaptation achieves the best results across all three tasks, improving over the Base Model by +1.5, +2.0, and +1.2, indicating

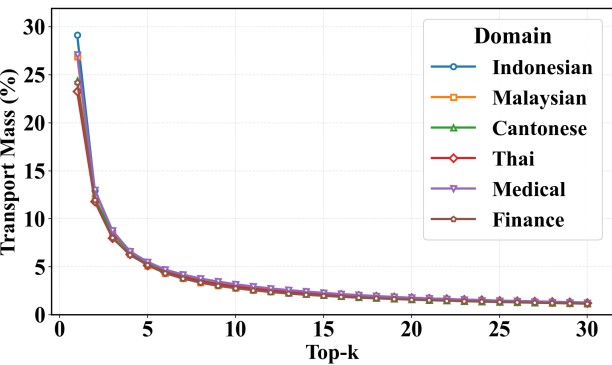

*Figure 3.* **Average transport mass explained** by the top-$k$ neuron correspondences, computed from the optimal transport plans and averaged over layers and modules.

*Table 7.* Effect of source model size on MalayMMLU. Results report accuracy (%) of Fused w/ Adaptation with a fixed 1B target model. Higher is better.

| Source Size | Humanities | Language | Others | STEM | Social Science |
|---|---|---|---|---|---|
| 1B | 47.65 | 41.76 | 47.97 | 46.05 | 46.00 |
| 8B | 48.81 | 41.68 | 48.69 | 46.58 | 46.91 |
| 32B | **49.17** | **42.27** | **49.17** | **46.75** | **47.28** |

SFT and distillation (Hinton et al., 2015) under normalized evaluation. Notably, the adapted fusion achieves the highest relative performance in every domain, whereas SFT and distillation show domain-dependent variability. These results suggest that effective cross-domain transfer requires not only shared representations, but also post-fusion adaptation to properly align domain-specific features. Details can be found in Appendix C.2.

### 5.4. Robustness of Cross-Architecture Merging

**Sensitivity to Source Model Backbone.** Figure 4 studies the sensitivity of our approach to the choice of high-resource source backbone. We compare our method (Fused w/ Adaptation) using two different source models, LLaMA3-8B and Qwen2.5-7B, while keeping the same low-resource target model. Across both Malay and Indonesian benchmarks, we observe consistent improvements over the Base Model. This suggests that our method transfers task-relevant features rather than overfitting to a specific source architecture, and remains robust across heterogeneous source models.

**Sensitivity to Fusion Coefficient $\alpha$.** Figure 5 analyzes the sensitivity of performance to the fusion coefficient $\alpha$, which controls the contribution of the source model during parameter merging. Across all categories, performance exhibits a consistent trend as $\alpha$ varies, indicating that our method is not overly sensitive to precise hyperparameter tuning. Moderate values of $\alpha$ (around 0.05–0.15) yield the strongest overall performance. However, when $\alpha$ is too small, the influence of the source model is limited, resulting in conservative

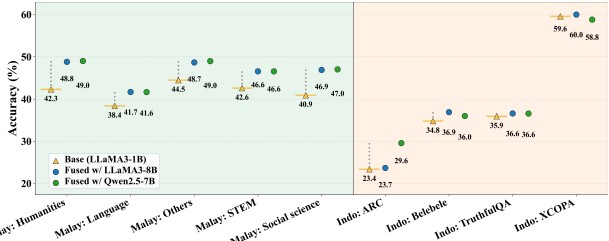

*Figure 4.* **Sensitivity to the choice of source backbone.** On the Malay and Indonesian benchmarks, our method consistently outperforms the Base Model when using either LLaMA3-8B or Qwen2.5-7B as the source model.

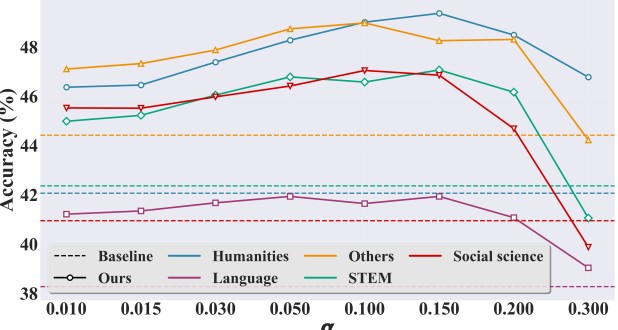

*Figure 5.* **Sensitivity analysis of the fusion coefficient $\alpha$** for our method (Fused w/ Adaptation) on MalayMMLU. We report category-wise accuracy (%; higher is better); dashed lines denote the corresponding base-model performance.

improvements. Conversely, overly large $\alpha$ values lead to performance degradation in several categories, suggesting that excessive source injection may introduce mismatched or noisy information under architectural differences.

**Impact on General Abilities.** Figure 6 evaluates how cross-architecture fusion affects general capabilities. Details of the general benchmarks used in this evaluation are provided in Appendix B.2. Overall, fusion does not degrade general abilities, and preserves performance close to the Base Model. For Malay and Cantonese, fusion with adaptation slightly improves the average general score, indicating that transferred knowledge can be integrated without harming

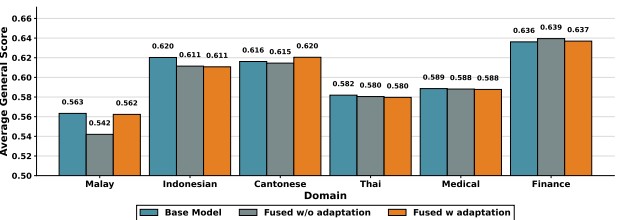

*Figure 6.* **Impact of cross-architecture fusion on general abilities across domains.** Bars show the average general score (higher is better) for the Base Model and fused variants, with and without post-fusion adaptation.

generalization. For Indonesian, Thai, and Medical settings, fused models achieve performance comparable to the Base Model, with differences within a narrow margin. These results suggest that our transport-based merging selectively injects transferable knowledge while largely maintaining the target model's original general capabilities.

## 6. Conclusion

We proposed an OT-based framework for cross-architecture model merging that aligns heterogeneous models in activation space and converts the resulting correspondences into direct weight-space fusion. Across low-resource languages and expert domains, our approach consistently improves target models using only a small calibration set, and residual-frozen adaptation further strengthens performance while preserving general capabilities. Overall, transport-guided cross-architecture merging provides a practical and principled alternative to distillation when architectures differ.

## Impact Statement

This work improves knowledge transfer from high-resource models to smaller low-resource or domain-specific models with heterogeneous architectures. We use activation-based optimal transport to estimate cross-architecture correspondences from a small input set for direct parameter fusion, optionally followed by lightweight adaptation. Positive impacts include reducing data/compute requirements for low-resource languages and specialized domains. Negative impacts include propagating biases, factual errors, or unsafe behaviors from the source model, and overstating robustness if evaluation is insufficient. We therefore emphasize rigorous evaluation, transparent reporting, and appropriate safeguards when applying the method in practice.

## Acknowledgements

This research/project is supported by the National Research Foundation, Singapore under its National Large Language Models Funding Initiative (AISG Award No: AISG-NMLP-2024-002). Any opinions, findings and conclusions or recommendations expressed in this material are those of the author(s) and do not reflect the views of National Research Foundation, Singapore.

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

# A. Optimal Transport Formulation Details

This appendix provides the full formulations underlying the transport plans used for model merging. We present both the neuron-level transport within each layer pair and the global layer-level mixing transport across the full depth.

## A.1. Neuron-Level Transport Polytope and Objective

To manage computation and filter out noise, we apply top-$k$ selection strategies at both the neuron and transport matrix levels.

**Transport Matrix Sparsification.** For a target layer $\ell$ with $n_\ell$ feature channels and a source layer $m$ with $n'_m$ feature channels, we define a cost matrix $C^{\ell m}_{\text{inner}} \in \mathbb{R}^{n_\ell \times n'_m}$, where each entry measures the dissimilarity between activation channels (Eq. 7).

We compute a soft relationship matrix $Q^{\ell m} \in \mathbb{R}^{n_\ell \times n'_m}$ by solving an entropically regularized optimal transport problem:

$$Q^{\ell m} = \arg \min_{Q \in \mathcal{T}(n_\ell, n'_m)} \langle C^{\ell m}_{\text{inner}}, Q \rangle - \varepsilon H(Q), \quad (23)$$

where $\varepsilon > 0$, $\langle A, B \rangle = \sum_{i,j} A_{ij} B_{ij}$, and the entropy term is

$$H(Q) = -\sum_{i=1}^{n_\ell} \sum_{j=1}^{n'_m} Q_{ij} \big( \log Q_{ij} - 1 \big). \quad (24)$$

The transportation polytope enforces balanced marginals:

$$\mathcal{T}(n_\ell, n'_m) = \Big\{ Q \in \mathbb{R}^{n_\ell \times n'_m}_+ : Q\mathbf{1} = a, \ Q^\top \mathbf{1} = b \Big\}, \quad (25)$$

where we use uniform marginals

$$a = \tfrac{1}{n_\ell} \mathbf{1} \in \mathbb{R}^{n_\ell}, \qquad b = \tfrac{1}{n'_m} \mathbf{1} \in \mathbb{R}^{n'_m}. \quad (26)$$

Uniform marginals ensure that each target channel distributes equal total mass and each source channel receives equal total mass, preventing degenerate, one-sided matchings.

## A.2. Layer-Level Transport for Global Mixing

Given neuron-level solutions $\{Q^{\ell m}\}$, we form the layer-level cost matrix $C_{\text{layer}} \in \mathbb{R}^{L \times M}$ by

$$C_{\text{layer}}[\ell, m] = \langle C^{\ell m}_{\text{inner}}, Q^{\ell m} \rangle. \quad (27)$$

We then compute the global layer mixing matrix $P \in \mathbb{R}^{L \times M}$ via

$$P = \arg \min_{P \in \mathcal{T}(L,M)} \langle C_{\text{layer}}, P \rangle - \eta H(P), \quad (28)$$

where $\eta > 0$ and

$$\mathcal{T}(L, M) = \Big\{ P \in \mathbb{R}^{L \times M}_+ : P\mathbf{1} = \bar{a}, \ P^\top \mathbf{1} = \bar{b} \Big\}, \quad (29)$$

with uniform marginals

$$\bar{a} = \tfrac{1}{L} \mathbf{1} \in \mathbb{R}^L, \qquad \bar{b} = \tfrac{1}{M} \mathbf{1} \in \mathbb{R}^M. \quad (30)$$

These constraints enforce global balance: each target layer allocates its mass across source layers, and each source layer receives comparable total mass, improving utilization during merging.

## A.3. Sinkhorn Solver and Stabilization

We solve the entropically regularized OT problems in Eqs. (23) and (28) using Sinkhorn iterations. Below we summarize the scaling-form derivation and the iterative updates.

### A.3.1. SCALING FORM

Consider the generic entropic OT problem

$$\min_{Q \in \mathbb{R}^{n \times m}_+} \langle C, Q \rangle - \varepsilon H(Q) \quad \text{s.t.} \quad Q\mathbf{1} = a, \ Q^\top \mathbf{1} = b. \quad (31)$$

Define the Gibbs kernel

$$K = \exp(-C/\varepsilon). \quad (32)$$

The optimal solution has the form

$$Q^\star = \text{diag}(u) \, K \, \text{diag}(v), \quad (33)$$

where $u \in \mathbb{R}^n_+$ and $v \in \mathbb{R}^m_+$ are scaling vectors chosen to match the marginals.

### A.3.2. SINKHORN ITERATIONS

Given $K$, Sinkhorn iterations alternately normalize rows and columns to satisfy the marginal constraints:

$$u \leftarrow a./(Kv), \qquad v \leftarrow b./(K^\top u), \quad (34)$$

where $./$ denotes element-wise division. After convergence, $Q = \text{diag}(u)K\text{diag}(v)$ satisfies $Q\mathbf{1} \approx a$ and $Q^\top \mathbf{1} \approx b$.

For the neuron-level problem, we use $C = C^{\ell m}_{\text{inner}}$, $a = \tfrac{1}{n_\ell}\mathbf{1}$, $b = \tfrac{1}{n'_m}\mathbf{1}$. For the layer-level problem, we use $C = C_{\text{layer}}$, $\bar{a} = \tfrac{1}{L}\mathbf{1}$, $\bar{b} = \tfrac{1}{M}\mathbf{1}$.

### A.3.3. PRACTICAL STABILIZATION

In practice, the kernel $K = \exp(-C/\varepsilon)$ may suffer from numerical underflow when costs are large or $\varepsilon$ is small. We therefore optionally employ standard stabilization strategies: (i) log-domain Sinkhorn updates, and/or (ii) periodic rescaling of $(u, v)$. We stop iterations when the maximum marginal violation $\max\{\|Q\mathbf{1} - a\|_\infty, \|Q^\top \mathbf{1} - b\|_\infty\}$ falls below a tolerance.

A.3.4. HYPERPARAMETER SETTINGS

We solve the entropically regularized OT problems using the Sinkhorn algorithm with the following specific hyperparameters:

**Inner OT (Feature Alignment).** For computing $Q^{\ell m}$ (Eq. (23)):

- Regularization ($\varepsilon$): We use $\varepsilon = 0.1$ for standard text datasets (e.g., Malay, Cantonese) and a tighter $\varepsilon = 0.03$ for the GSM8K math reasoning task.

- Iterations: We use a memory-efficient streaming Sinkhorn solver with fixed 200 iterations.

- Tolerance: Convergence tolerance is set to $10^{-6}$.

**Outer OT (Layer Mixing).** For computing $P$ (Eq. (28)):

- Regularization ($\eta$): We set the default $\eta = 0.1$. We employ an adaptive scaling mechanism where $\eta$ is increased if the maximum value of the cost matrix $C_{\text{layer}}$ exceeds 1000, ensuring numerical stability.

- Iterations: We allow up to 1000 iterations.

- Tolerance: We use a strict convergence tolerance of $10^{-9}$ with a numerical stability epsilon of $10^{-12}$.

### A.4. Proofs for Representation-Space Interpretation

*Proof of Theorem 4.1.* By definition,

$$\widetilde{W}_{B \to A}^{\ell m} h_A = (\Phi_{\text{out}}^{\ell m} W_B^m \Phi_{\text{in}}^{\ell m}) h_A = \Phi_{\text{out}}^{\ell m}\big(W_B^m(\Phi_{\text{in}}^{\ell m} h_A)\big),$$

which is exactly Eq. (21). $\qquad\square$

*Proof of Corollary 4.2.* Assume $\Phi_{\text{in}}^{\ell m}$ and $\Phi_{\text{out}}^{\ell m}$ are invertible. Let $U : \mathbb{R}^{d_{A,\text{in}}^\ell} \to \mathbb{R}^{d_{A,\text{out}}^\ell}$ be any linear map such that

$$U h_A = \Phi_{\text{out}}^{\ell m}\Big(W_B^m\big(\Phi_{\text{in}}^{\ell m} h_A\big)\Big) \quad \text{for all } h_A.$$

Then

$$U = \Phi_{\text{out}}^{\ell m} W_B^m \Phi_{\text{in}}^{\ell m} = \widetilde{W}_{B \to A}^{\ell m},$$

establishing uniqueness. $\qquad\square$

### A.5. Neuron Selection for Replacement.

For the top-$k$ neuron replacement strategy, we set the default number of neurons to $k = 128$. The selection rule matches the main text: we run the target model on the calibration set $\mathcal{D}$ (used for transport estimation) and record neuron activations via forward hooks. For each neuron $j$ in layer $\ell$, we compute its activation strength as the mean absolute activation over the $T$ samples,

$$s_j = \frac{1}{T} \sum_{t=1}^{T} |h_{t,j}|,$$

where $h_{t,j}$ denotes the activation of neuron $j$ on sample $t$. We then select the indices of the $k$ neurons with the highest $s_j$.

### A.6. Computational Complexity

We analyze the computational cost of the proposed optimal transport (OT)–based alignment in terms of feature- and layer-level transport.

**Feature-level OT.** For each target layer $\ell$ and source layer $m$, we solve an entropically regularized OT problem between $n_\ell$ target features and $n_m'$ source features using Sinkhorn iterations. Each Sinkhorn update involves matrix–vector multiplications with the kernel $K^{\ell m} \in \mathbb{R}^{n_\ell \times n_m'}$, incurring $\mathcal{O}(n_\ell n_m')$ time per iteration. With $I_{\text{in}}$ Sinkhorn iterations, the total cost of one inner OT problem is $\mathcal{O}(I_{\text{in}} n_\ell n_m')$.

Across all layer pairs, the total feature-level OT cost is

$$\mathcal{O}\left(I_{\text{in}} \sum_{\ell=1}^{L} \sum_{m=1}^{M} n_\ell n_m'\right).$$

**Layer-level OT.** The layer-level transport problem operates on a cost matrix $C_{\text{layer}} \in \mathbb{R}^{L \times M}$. Each Sinkhorn iteration costs $\mathcal{O}(LM)$, and with $I_{\text{out}}$ iterations, the total cost is $\mathcal{O}(I_{\text{out}} LM)$, which is negligible compared to feature-level OT when $n_\ell, n_m' \gg L, M$.

**Overall complexity and practicality.** In practice, $I_{\text{in}}$ and $I_{\text{out}}$ denote the numbers of Sinkhorn iterations for the inner and outer OT problems (not to be confused with $T$, the number of samples used for activation extraction in Eq. (6)). We fix $I_{\text{in}}$ and $I_{\text{out}}$ to small constants (e.g., 200 and 1000), and feature dimensions are moderate for selected layers and projection modules. Moreover, OT estimation is performed *once* using a small dataset and does not introduce overhead during inference. As a result, the OT-based alignment adds a one-time preprocessing cost and remains practical for cross-architecture fusion.

## B. Experimental Setup

In this section, we detail the datasets used for training our models and the comprehensive benchmarks employed to evaluate their performance across diverse domains and languages.

## B.1. Stimulus and Training Datasets

We use stimulus datasets to extract representative activations for correspondence estimation, and training datasets for task-specific adaptation when required. Both are tailored to each domain and language. For each dataset, we randomly sample 2000 examples.

### B.1.1. DOMAIN-SPECIFIC

- **Medical LLaMA3:** We employ a specialized medical dataset (medical_LLaMA3) (Shekswess, 2024) containing high-quality biomedical literature, clinical case reports, and medical guidelines to instill domain-specific medical knowledge.

- **Finance:** To cover the financial domain, we utilize a dedicated finance dataset (finance) (Flowers, 2025) comprising financial news, reports, and economic analysis texts, enabling the model to grasp complex economic concepts and terminology.

### B.1.2. MULTILINGUAL LANGUAGES

- **Thai:** We utilize fineweb_thai (Penedo et al., 2024), a subset of the FineWeb corpus focused on high-quality Thai web text, to improve language modeling performance in Thai.

- **Indonesian:** We incorporate indonesian_conversation, a dataset designed to capture natural dialogue and conversational nuances in the Indonesian language.

- **Malay:** For the Malay language, we use malaysian_sft (Mesolitica, 2024), a supervised adaptation dataset tailored to improve instruction following in Malaysian contexts.

- **Cantonese:** We include a dedicated cantonese dataset (Lynn, 2024) to address the linguistic specificities and colloquialisms of this regional language.

## B.2. Evaluation Benchmarks

We evaluate our model on a diverse set of benchmarks, categorized by language and domain to provide a holistic view of performance.

### B.2.1. MALAY BENCHMARKS

- **MMLU (Malay):** To assess general world knowledge in Malay, we utilize the Malay version of the Massive Multitask Language Understanding (MMLU) benchmark (Poh et al., 2024). This task evaluates the model's accuracy in a zero-shot setting across a wide range of subjects—including STEM, humanities, and social sciences—adapted to the Malay language to test cross-lingual knowledge transfer.

### B.2.2. INDONESIAN BENCHMARKS

- **ARC (Indonesian):** The AI2 Reasoning Challenge (ARC) (Clark et al., 2018) consists of grade-school science questions designed to test complex reasoning and scientific knowledge. We utilize the Indonesian version to evaluate the model's reasoning capabilities in a low-resource language context.

- **Belebele (Indonesian):** Belebele (Bandarkar et al., 2024) is a multilingual machine reading comprehension (MRC) dataset. Based on the FLORES-200 passages, it evaluates the model's ability to answer multiple-choice questions given a specific context. We report results on the Indonesian (ind_Latn) subset.

- **TruthfulQA-MC (Indonesian):** TruthfulQA (Lin et al., 2022) evaluates the model's truthfulness and its tendency to generate imitative falsehoods. We use the Indonesian translated version and report the average accuracy across the single-true (MC1) and multi-true (MC2) multiple-choice tasks.

- **XCOPA (Indonesian):** The Cross-lingual Choice of Plausible Alternatives (XCOPA) (Ponti et al., 2020) assesses causal commonsense reasoning. The task requires the model to identify the correct cause or effect given a premise. We evaluate performance on the Indonesian subset (xcopa_id).

### B.2.3. CANTONESE BENCHMARKS

- **CMMLU (Cantonese):** We evaluate the model's proficiency in the Cantonese (Yue) dialect using the CMMLU dataset from the Yue-Benchmark suite (Jiang et al., 2025). This benchmark consists of multiple-choice questions covering diverse disciplines such as history, physics, and culture. It is specifically designed to test the model's ability to understand and reason with Cantonese-specific vocabulary, grammar, and colloquialisms.

### B.2.4. THAI BENCHMARKS

- **MMLU (Thai):** We assess general world knowledge in Thai from the MMLU benchmark (Hendrycks et al., 2020) (adapted as mmlu_prox_lite_th_other). This subset comprises diverse multiple-choice questions across various uncategorized domains, evaluating the model's breadth of knowledge beyond specific academic disciplines in the Thai language.

- **XCOPA (Thai):** To evaluate causal reasoning in Thai, we utilize the Thai subset (xcopa_th) of the XCOPA benchmark (Ponti et al., 2020). Similar to the Indonesian task, the model must select the most plausible alternative between cause and effect based on a premise.

- **MGSM (Thai):** The Multilingual Grade School Math (MGSM) benchmark (Shi et al., 2022) evaluates arithmetic reasoning capabilities. We employ the Chain-of-Thought (CoT) prompting setting on the Thai subset (`mgsm_cot`) to assess the model's ability to perform multi-step mathematical reasoning in Thai.

### B.2.5. FINANCE BENCHMARKS

To assess the model's capabilities in the finance and business domains, we utilize three relevant subsets from the MMLU benchmark (Hendrycks et al., 2020):

- **MMLU (Business Ethics):** This task (`global_mmlu_full_en_business_ethics`) evaluates the model's ability to apply ethical principles in business scenarios, covering topics such as corporate governance, moral reasoning, and professional standards.

- **MMLU (Microeconomics):** This subset (`global_mmlu_full_en_high_school_microeconomics`) covers fundamental economic concepts including supply and demand, market structures, and consumer behavior, reflecting a high school level understanding of microeconomic theory.

- **MMLU (Professional Accounting):** Designed to test advanced accounting knowledge, this task (`global_mmlu_full_en_professional_accounting`) includes questions on financial accounting, reporting standards, and auditing, corresponding to professional certification levels.

### B.2.6. MEDICAL BENCHMARKS

We evaluate domain-specific medical knowledge using three subsets from the Massive Multitask Language Understanding (MMLU) benchmark (Hendrycks et al., 2020):

- **MMLU (Anatomy):** This task tests the model's knowledge of human anatomy through multiple-choice questions covering various body systems and structures, derived from academic and professional sources.

- **MMLU (Medical Genetics):** This subset evaluates the model's understanding of genetic principles, hereditary diseases, and clinical genetics, requiring specialized medical knowledge to answer correctly.

- **MMLU (Professional Medicine):** This task assesses the model's ability to apply medical knowledge in professional contexts. It includes questions typical of medical board examinations, covering diagnosis, treatment, and clinical decision-making.

### B.3. General Capabilities

To assess the model's fundamental reasoning and commonsense abilities independent of specific domains or languages, we employ the following benchmarks:

- **ARC-Easy (Clark et al., 2018):** The AI2 Reasoning Challenge (Easy Set) consists of grade-school science questions. It is designed to test the model's ability to answer questions that require basic commonsense reasoning and world knowledge.

- **CommonsenseQA (Talmor et al., 2019):** This dataset evaluates commonsense reasoning through multiple-choice questions that require prior knowledge to distinguish between plausible answers, challenging the model's understanding of semantic relationships.

- **PIQA (Bisk et al., 2020):** The Physical Interaction QA (PIQA) benchmark focuses on physical commonsense reasoning. It presents the model with everyday situations and asks it to predict the most plausible physical outcome or interaction.

- **Social IQA (Sap et al., 2019):** To evaluate social intelligence, we use Social IQA, which tests the model's ability to reason about social interactions, including understanding motivations, emotional reactions, and likely next steps in social contexts.

- **WinoGrande (Sakaguchi et al., 2021):** This benchmark is a large-scale dataset for commonsense reasoning, formulated as fill-in-the-blank problems. It is designed to be more robust against annotation artifacts than the original Winograd Schema Challenge, requiring deep understanding to resolve ambiguous pronouns.

## C. Details about Experiments

### C.1. Visualization of Feature-Level Transport Maps

We provide qualitative visualizations of feature-level optimal transport maps to further illustrate cross-architecture representation similarity across heterogeneous architectures. Figure 7 and Figure 8 visualize representative transport plans $Q^{\ell m}$ computed between target and source layers.

Each entry of $Q^{\ell m}$ encodes the amount of transport mass between a target neuron (horizontal axis) and a source neuron (vertical axis), where brighter values indicate stronger cross-architecture feature similarity. Across both language settings, the transport mass is highly non-uniform and concentrates on sparse, localized regions rather than being evenly distributed. Only a small subset of neuron pairs exhibits strong transport values, while the majority of entries remain close to zero.

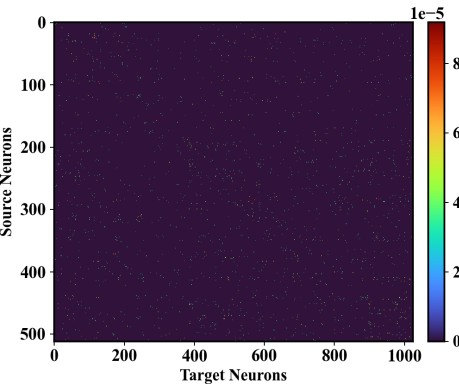

*Figure 7.* Feature-level optimal transport map between a LLaMA-3-1B model and a Chinese LLaMA-8B model at layer 5 (`K` projection). Brighter values indicate stronger neuron-level alignment.

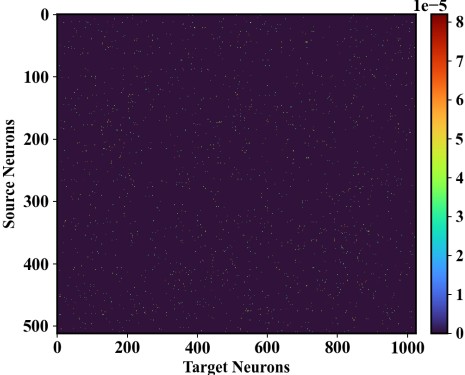

*Figure 8.* Feature-level optimal transport map between a Malaysian-version LLaMA-3-1B model and a LLaMA-8B model at layer 0 (`K` projection). Similar to Cantonese, transport mass concentrates on sparse, high-similarity neuron pairs.

These sparse yet structured patterns indicate that heterogeneous language models share well-aligned internal representations at the neuron level, rather than exhibiting random or diffuse correspondence. Such visual evidence complements our quantitative analysis and supports the design choice of top-$k$ neuron replacement, as the most informative cross-architecture alignments are concentrated in a small number of highly similar neuron pairs.

### C.2. Details about Comparison with SFT training and Distillation.

**Evaluation Protocol and Normalization.** To enable a fair comparison across domains with heterogeneous score ranges, we normalize results on a per-domain basis. Specifically, let $x_{d,m}$ denote the raw performance of method $m$ on domain $d$. We apply two complementary normalization schemes.

**Z-score normalization.** For each domain $d$, we compute

$$z_{d,m} = \frac{x_{d,m} - \mu_d}{\sigma_d},$$

where $\mu_d$ and $\sigma_d$ are the mean and standard deviation of scores across all methods for domain $d$. This normalization highlights relative performance within each domain and is used to visualize method-wise trajectories and average trends. Z-scores are shown in the top row (domain-specific trajectories) and the bottom-right panel (average across domains).

**Min–max scaling.** We additionally apply min–max normalization per domain,

$$\hat{x}_{d,m} = \frac{x_{d,m} - \min_d}{\max_d - \min_d},$$

where $\min_d$ and $\max_d$ denote the minimum and maximum scores across methods for domain $d$. This scaling preserves the relative shape of improvements within each domain and is used in the bottom-left panel to compare improvement patterns across domains.

**Figure Interpretation.** The top row of Figure 1 shows z-score–normalized performance trajectories for representative domains (Cantonese, Malay, and Thai), illustrating how different training strategies compare within each domain. The bottom-left panel visualizes min–max–scaled scores across all domains, highlighting the relative improvement patterns of each method. The bottom-right panel reports the average z-score across domains, summarizing overall cross-domain performance.

Together, these normalized views demonstrate that our adapted merging method consistently achieves the strongest relative performance across domains, while SFT and distillation exhibit more domain-dependent variability.

## D. Additional Discussion

**Q1: How sensitive is the method to the choice of the calibration set $\mathcal{D}$?** Our framework estimates transport plans $\{Q^{\ell m}\}$ and $P$ from activation statistics, and therefore relies on $\mathcal{D}$ to elicit representative behaviors of the target task or domain. If $\mathcal{D}$ is severely mismatched, the resulting correspondences may become noisy and less informative. To reduce this sensitivity, our design explicitly incorporates several stabilizing choices: (i) using a moderately sized but task-relevant calibration set, (ii) employing entropic regularization in OT to smooth noisy similarity estimates, and (iii) restricting fusion to a small set of top-$k$ highly activated neurons to limit interference. In practice, we find that lightweight data from the target language or domain is usually sufficient. When no such data is available, approaches that rely on explicit supervision (e.g., distillation or adapters) may be more appropriate for that setting.

**Q2: Under what conditions can cross-architecture fusion fail despite sharp OT correspondences?** Even when OT yields seemingly confident correspondences, effective fusion is not guaranteed if the target model lacks sufficient capacity to absorb the injected knowledge, or if the transferred features are misaligned with the target task. These cases reflect fundamental capacity or task-mismatch constraints rather than failures of the transport mechanism itself. Our method adopts a conservative strategy—top-$k$ neuron replacement combined with optional residual-frozen adaptation—to mitigate such risks by limiting the scope of transfer and allowing controlled recalibration of the remaining parameters.

**Q3: Can fusion propagate undesirable behaviors or biases from the source model?** As with any direct parameter transfer method, our approach may propagate both beneficial capabilities and undesirable behaviors present in the source model. This is an inherent consideration of weight-level reuse rather than a property unique to our framework. In practical deployments, we recommend applying standard post-hoc safety and alignment pipelines (e.g., safety evaluation, filtering, or targeted alignment) to the fused model. Incorporating safety-aware objectives directly into the transport formulation is an interesting direction for future work.

