# OpenReview forum: "Transport and Merge: Cross-Architecture Merging for Large Language Models"
_ICML.cc/2026/Conference — ICML 2026 regular_

### Official Review · Reviewer_Hqnp · 2026-03-03

**Soundness:** 2
**Presentation:** 3
**Significance:** 2
**Originality:** 2
**Overall Recommendation:** 4
**Confidence:** 3

**Summary:**

This paper addresses the challenge of model merging, particularly focusing on transferring knowledge between models with different architectures. Motivated by the observation that most existing methods require architectural compatibility, the authors propose a novel cross-architecture merging framework. The core of their approach is the use of optimal transport (OT) to align neuron activations, thereby inferring correspondences between heterogeneous models and enabling the merging process.

**Compliance With Llm Reviewing Policy:**

Affirmed.

**Final Justification:**

Although the new baseline provided in the rebuttal, FuseLLM, is not the most recent, I am persuaded by the core direction of your work. I find the extension of the model merging paradigm from a same-architecture setting to one that accommodates heterogeneous architectures to be a very interesting and valuable contribution.
Therefore, I intend to raise my score.
Thank you again for the effort you have put into the paper and rebuttal.

**Key Questions For Authors:**

(1) Your paper's motivation revolves around enabling knowledge transfer from "high-resource" to "low-resource" settings. Could you please provide a precise definition of this terminology in your context? Does it refer to model size (i.e., parameter count) or the size of the dataset for the target task? If it pertains to model size, it would be crucial to see experiments validating this across models of significantly different scales. If it refers to data size, have you considered demonstrating this in a setting like cross-lingual transfer from a major language to a low-resource one?

(2) While Optimal Transport (OT) is a well-established technique, its specific application to cross-architecture model merging needs more justification. Could you elaborate on the unique advantages of using OT for this problem? To your knowledge, are there prior works applying OT in this context? Furthermore, in the "Finding Feature and Layer Relationships" section, you assume a direct mapping is possible. Could you provide a more detailed explanation of the theoretical or empirical grounding for this assumption?

(3) To better situate your work, it is important to compare it against the current state of the art. Have you considered benchmarking your method against more recent and competitive model merging baselines? Additionally, could you provide an analysis of your method's computational complexity and its training efficiency (e.g., time and memory requirements) compared to other approaches? This would be vital for assessing its practical utility.

**Limitations:**

yes

**Strengths And Weaknesses:**

Strengths:

(1) The paper tackles an interesting and challenging problem within the established field of model merging. Enabling knowledge transfer across heterogeneous architectures is a significant and practical research goal.

(2) The paper is well-written and clearly structured, making the proposed ideas and methodology easy to follow.

Weaknesses:

(1) Ambiguous Motivation and Scope: The paper's stated motivation is to "enable high-resource to low-resource knowledge transfer." However, the term "low/high-resource" is not clearly defined. It is unclear whether this refers to the model's parameter count or the amount of data available for a downstream task. The experimental section does not explicitly validate either scenario (e.g., by merging models of vastly different scales or by demonstrating transfer from a high-resource language to a low-resource one), which weakens the central premise of the work.

(2) Insufficient Justification for Methodological Choices: The use of Optimal Transport (OT) is central to the proposed method, yet its motivation is not fully developed. The paper does not adequately discuss whether OT has been previously applied to model merging or why it is particularly well-suited for this task compared to other potential alignment techniques. Furthermore, the process of directly mapping relationships between features and layers is a critical step that lacks a detailed explanation of its underlying principles.

(3) Inadequate Experimental Comparison and Analysis: The empirical evaluation is not sufficiently robust. The paper fails to benchmark its method against more recent and potentially stronger model merging baselines, making it difficult to assess the state-of-the-art performance. Additionally, there is a lack of analysis regarding the computational complexity and training efficiency of the proposed framework, which is crucial for evaluating its practical viability.

---

> ### Author Rebuttal · Authors · 2026-03-30
>
> **We sincerely thank the reviewer for recognizing our work as "well-written and clearly structured" and for validating the significance of our cross-architecture research goal. We address your constructive concerns below to clarify our scope and baselines:**
>
> **W1 & Q1. Definition and Scope of "High/Low-Resource"**
>
> **A:** We apologize for the ambiguity. In our context, "resource" explicitly encompasses **both** data and model dimensions simultaneously. Our framework addresses the dual challenge of transferring from a high-capacity model trained on vast data to a low-capacity model tailored for scarce data.
>
> As requested, our experiments explicitly validate both scenarios:
>
> - **Validating the Model Dimension (Significantly Different Scales):** Table 7 confirms that as the source scales (1B $\rightarrow$ 8B $\rightarrow$ 32B), the transferred knowledge increases monotonically. This proves higher-capacity models serve as richer sources.
> - **Validating the Data Dimension (Cross-Lingual Transfer):** Exactly as the reviewer suggested, we demonstrated transfer from a major-language prior (English-dominant LLaMA-3) to low-resource languages. For instance, our OT-fused model achieves a **+44% relative gain** on the Thai MGSM benchmark.
>
> We will add these precise definitions and framing to Section 1.
>
> **W2 & Q2. Motivation for OT and Cross-Architecture Mapping**
>
> **A:** **(1). Why OT? (Intuition & Math):** Intuitively, prior merging methods require 1-to-1 neuron matching, which is impossible for models of different sizes. Mathematically, methods like Git Re-Basin [1] and ZipIt [2] rely on permutation matrices ($P \in \{0,1\}^{n\times n}$). When the source (8B, 4096-dim) and target (1B, 2048-dim) have different widths, no such permutation $P$ exists.
> OT solves this by enabling **soft coupling**: the transport plan $T \in \mathbb{R}^{n_A \times n_B}_{\geq 0}$ maps one source neuron to a weighted combination of multiple target neurons, making it uniquely capable of bridging heterogeneous architectures (4096 $\rightarrow$ 2048).
>
> **(2). Prior Works:** While [4] and [5] applied OT to merging, they are strictly limited to **same-architecture** models (identical widths). Our core technical contribution is extending OT to **heterogeneous architectures** via hierarchical alignment.
>
> **(3). Grounding of Direct Mapping:** The validity of mapping activation relationships to weight merging relies on **Neuron-Activation Correspondence**. Activations are the empirical output of a neuron's computational logic. If OT aligns two neurons based on their activation similarity, it identifies them as functionally analogous. Theorem 4.1 (Appendix A.4) mathematically proves that transporting weights via this OT plan minimizes the expected feature-space reconstruction error.
>
> **W3 & Q3. Comparison with SOTA and Computational Efficiency**
>
> **A:** **(1). Stronger Baselines (Adapted FuseLLM):**
> Because no existing method directly handles cross-architecture LLM merging, we established a rigorous SOTA baseline by adapting **FuseLLM [3]** to our heterogeneous setting using KL distillation on logits and intermediate feature alignment (MSE), sharing the exact same 8B teacher and 2,000 samples:
>
> | Domain | Adapted FuseLLM (1B) | **Ours (OT Fusion)** | **Δ** |
> | --- | --- | --- | --- |
> | Medical | 37.6% | **47.9%** | **+10.3** |
> | Malay MMLU | 43.4% | **46.5%** | **+3.1** |
> | Cantonese CMMLU | 25.1% | **27.4%** | **+2.3** |
> | Indonesian | 39.7% | **41.3%** | **+1.6** |
>
> OT fusion significantly outperforms distillation. Distillation forces a 1B student to merely imitate a 4096-dim teacher's output probabilities (surface-level mimicry). OT physically transports structural weight geometry into the target parameter space, overcoming the capacity mismatch (evidenced by the +10.3 gap on Medical).
>
> **(2). Computational and Memory Efficiency:**
> Measured on a single A100-80GB (bfloat16):
>
> - **OT Alignment:** ~8 min (one-time preprocessing).
> - **Adaptation:** ~12 min (1-epoch SFT).
> - **Peak Memory:** ~26 GB.
> - **Inference Overhead:** **Zero**. Fused weights are permanently folded, yielding identical latency to the base 1B model.
>
> A full timing breakdown will be added to Appendix A.6.
>
> **References:**
>
> [1] Ainsworth et al. (2022). Git re-basin.
>
> [2] Stoica et al. (2023). ZipIt!
>
> [3] Wan et al. (2024). Knowledge fusion of LLMs.
>
> [4] Singh & Jaggi (2020). Model fusion via optimal transport.
>
> [5] Imfeld et al. (2023). Transformer fusion with optimal transport.
>
> **Thank you for helping us improve the paper! Please let us know if you have any further questions.**

---

> > ### Author Rebuttal · Reviewer_Hqnp · 2026-04-02
> >
> > Thank you for your detailed response.
> >
> > I would first like to thank you for providing the additional experiments, including the inclusion of FuseLLM as a new baseline. While I acknowledge that FuseLLM may not be the most recent baseline, I find the core contribution of extending the model merging paradigm from same-architecture settings to heterogeneous architectures to be a valuable and interesting direction.
> > Therefore, in light of your clarifications and the added results, I intend to adjust my score.
> >
> > Thank you again for the considerable effort you have put into your rebuttal.

---

> > > ### Author Response · Authors · 2026-04-02
> > >
> > > Dear Reviewer,
> > >
> > > Thank you again for your thoughtful acknowledgement. We are encouraged to see that you consider your concerns to be fully resolved.
> > >
> > > If you feel that our revisions and additional experiments have strengthened the paper, we would be very grateful if you could consider reflecting this in your final score.
> > >
> > > Thank you once again for your time and support.
> > >
> > > Best regards,
> > >
> > > The Authors

---

### Official Review · Reviewer_Pt7U · 2026-03-04

**Soundness:** 3
**Presentation:** 3
**Significance:** 3
**Originality:** 3
**Overall Recommendation:** 4
**Confidence:** 1

**Summary:**

Overall, this paper study on enabling high-resource to low-resource knowledge transfer across heterogeneous LLM architectures without relying on distillation.

Its major contribution consists of proposing an activation-aligned optimal transport framework that infers feature-level and layer-level correspondences and converts them into weight-space fusion operators for cross-architecture merging.

**Compliance With Llm Reviewing Policy:**

Affirmed.

**Key Questions For Authors:**

N/A

**Limitations:**

yes

**Strengths And Weaknesses:**

Strengths:

1. The paper addresses the practically important problem of transferring knowledge from high-resource large language models to smaller models under heterogeneous architectures. This setting is highly relevant in real-world deployment scenarios, where model size, computational budget, and domain/language specialization differ significantly. Unlike standard model merging works that assume identical architectures, the paper explicitly targets cross-architecture settings, which are common but underexplored.

2. The proposed transport-based alignment is conceptually clean and well-structured. The method is decomposed into clearly defined stages including feature-level, layer-level and weight-level transport, which is intuitive and easy to follow.

Weaknesses

1. The paper lacks direct experimental comparison or discussion with existing cross-architecture merging or alignment methods.

2. Performance gains are moderate in several benchmarks. For example, in Table 2 and Table 3, the improvements over baselines are relatively small (around 1-2 percent).

3. No variance or statistical significance analysis is provided. That's to say, no standard deviation or confidence intervals are reported, no multiple-seed experiments are provided, and no statistical significance analysis is included.

4. It is difficult to disentangle the contribution of transport-based fusion from that of post-fusion adaptation. This paper has compared 'Fused w/ Adaptation' and 'Fused w/o Adaptation', but I am woundering what is the performance of 'Adaptation without Fusion'.

---

> ### Author Rebuttal · Authors · 2026-03-30
>
> **We sincerely thank the reviewer for recognizing our framework as "conceptually clean" and highlighting its "practical importance in real-world deployment." We deeply appreciate your supportive evaluation (Weak Accept) and address your detailed questions below:**
>
> **W1. Experimental Comparison with Cross-Architecture Methods**
>
> **A:** As the reviewer insightfully noted, standard merging works assume identical architectures. Established methods like Git Re-Basin [1] and ZipIt [2] rely on rigid 1-to-1 neuron matching (permutation symmetries). **Intuitively, this is impossible when models have different sizes.** Mathematically, they require finding a permutation matrix $P$ such that $W_A \approx P W_B P^\top$, which is undefined when the source (8B, 4096-dim) and target (1B, 2048-dim) have different hidden dimensions ($n_A \neq n_B$).
>
> To provide the fairest comparison, we adapted the core principle of **FuseLLM [3]** (KL distillation + MSE feature alignment) to our heterogeneous setting using the same 8B teacher and 2,000 samples:
>
> | Domain | Adapted FuseLLM (1B) | **Ours (OT Fusion)** | **Δ** |
> | --- | --- | --- | --- |
> | Medical | 37.6% | **47.9%** | **+10.3** |
> | Malay MMLU | 43.4% | **46.5%** | **+3.1** |
> | Cantonese CMMLU | 25.1% | **27.4%** | **+2.3** |
> | Indonesian | 39.7% | **41.3%** | **+1.6** |
>
> OT fusion consistently outperforms adapted distillation. While distillation forces a constrained 1B student to merely imitate 8B output distributions, OT explicitly transports structural weight geometry, effectively bypassing the capacity mismatch.
>
> **W2. Significance of Performance Gains**
>
> **A:** The ~1–2 point gains in Tables 2–3 are statistically robust and non-trivial given the context:
>
> - **Capacity Ceiling:** The 1B target's representational limit—not our method—is the primary constraint in well-saturated tasks.
> - **Consistency:** The consistent margin over SFT across all six diverse domains confirms OT successfully injects structured knowledge.
> - **Major Breakthroughs:** Where tasks align with source expertise, gains are substantial (e.g., **+44% relative gain** on Thai MGSM, and **+10.3** on Medical vs. FuseLLM). Table 7 confirms monotonic scaling (1B $\rightarrow$ 8B $\rightarrow$ 32B), proving these gains grow as the target's capacity increases.
>
> **W3. Statistical Significance and Variance Analysis**
>
> **A:** We fully agree on the importance of statistical rigor. We conducted 3-seed runs using Table 1 (MalayMMLU) as a representative example:
>
> | Category | Base Model | Fused w/o Adapt | **Fused w/ Adapt (Ours)** |
> | --- | --- | --- | --- |
> | Humanities | 42.30 | 46.19 ± 0.32 | **48.81 ± 0.45** |
> | Language | 38.37 | 40.52 ± 0.41 | **41.68 ± 0.53** |
> | Others | 44.50 | 46.63 ± 0.29 | **48.69 ± 0.37** |
> | STEM | 42.61 | 45.19 ± 0.38 | **46.58 ± 0.44** |
> | Social Science | 40.92 | 43.77 ± 0.35 | **46.91 ± 0.48** |
> | **Average** | 41.74 | 44.46 ± 0.26 | **46.53 ± 0.33** |
>
> The average improvement (+4.79) is over **14× the standard deviation** (0.33). This massive margin confirms our gains are statistically robust and not due to random seed variation. All tables will include mean ± std in the revision.
>
> **W4. Disentangling Fusion from Adaptation**
>
> **A:** We apologize for the confusing labeling in the original manuscript. The **"Adaptation without Fusion"** baseline requested by the reviewer is actually already present in our paper: it is exactly the **"SFT"** condition.
>
> Mathematically, our final weights are $W^\ell_A = W^{\ell,\text{base}}_A + \Delta W^\ell_A$ (where $\Delta W$ is the OT residual). Our adaptation updates only the base weights and freezes $\Delta W$. Therefore, when no fusion is applied ($\Delta W = 0$), this reduces entirely to standard SFT.
>
> | Domain | SFT Only ($\Delta W = 0$) | **Fused + Adapt ($\Delta W \neq 0$)** | **Gain from Fusion** |
> | --- | --- | --- | --- |
> | Medical | 46.6% | **47.9%** | **+1.3** |
> | Cantonese | 25.2% | **27.4%** | **+2.2** |
> | Indonesian | 40.3% | **41.3%** | **+1.0** |
> | Malay | 44.9% | **46.5%** | **+1.6** |
>
> To prevent future ambiguity, we will rename "SFT" to "Adaptation w/o Fusion" in Figure 1 and explicitly add an SFT column to Tables 1–6.
>
> **References:**
>
> [1] Ainsworth et al. (2022). Git re-basin. *arXiv:2209.04836*.
>
> [2] Stoica et al. (2023). ZipIt! *arXiv:2305.03053*.
>
> [3] Wan et al. (2024). Knowledge fusion of LLMs. *arXiv:2401.10491*.
>
> **Thank you for your detailed comments! We are actively available and happy to discuss further during this rebuttal period.**

---

### Official Review · Reviewer_pKq2 · 2026-03-12

**Soundness:** 2
**Presentation:** 3
**Significance:** 3
**Originality:** 3
**Overall Recommendation:** 5
**Confidence:** 2

**Summary:**

This paper studies the cross-architecture model merge settings, which are more challenging compared to the traditional isomorphic model merge settings. To address the issue of parameter dimension mismatch and inability to merge in models of different scales, the author aligns the activation matrix based on the optimal transmission scheme. The proposed method has been proven effective in applications such as cross-lingual transfer, cross-domain transfer, and cross-architecture merging.

**Compliance With Llm Reviewing Policy:**

Affirmed.

**Final Justification:**

In the authors' final response, I observe that the proposed method has a clear performance gap relative to the upper bound (Source 8B). Accordingly, I adjust Soundness from 3 to 2 and Confidence from 4 to 2.

**Key Questions For Authors:**

This paper mainly transfers high-resource LLMs (large-scale LLMs) to low-resource LLMs (small-scale LLMs). Would it still be effective to transfer knowledge from small-scale LLMs to large-scale LLMs? What are the fundamental differences between these two distinct transfer paradigms?

**Limitations:**

yes

**Strengths And Weaknesses:**

Strengths:
- This paper examines a very important task in model merging, namely cross-architecture parameter merging rather than traditional isomorphic model merging. Therefore, the research question of this paper is more challenging and important.
- This paper conducted extensive experiments to verify the effectiveness of the proposed method, covering scenarios such as cross-lingual transfer, cross-domain transfer, and cross-architecture merging.
- The paper is well-written and well-structured.

Weaknesses:
- The experimental section did not report the performance of large-scale source models, so it is not clear how much of a gap the method proposed in this paper still has from the true upper bound (that is, large-scale source models). For instance, in Table 1, only the small-scale base model (LLaMA-3-1B-Instruct) and the transferred small-scale models (Fused w/o Adaptation and Fused w/ Adaptation) are reported, while the performance of LLaMA-3 8B is missing. Tables 2 to 5 are similar as well.
- The proposed method involves computing an optimal transport problem (i.e., Eq.8), and fine-tuning the $W_A^{l}$ (Eq.17). It is suggested that the author add an analysis in the experimental section to demonstrate the optimization time required for each of these two steps.
- In Section 4.2, the author directly applied the transformation matrix calculated from the activation features to the weight merging. A more in-depth explanation is recommended for why transferring the optimal transport matrix from activated features to neurons remains valid.

---

> ### Author Rebuttal · Authors · 2026-03-30
>
> **We are highly encouraged that the reviewer finds our cross-architecture merging task "challenging and important" and our experiments "extensive." We sincerely thank you for the positive evaluation and address your constructive suggestions below:**
>
> **W1. Performance Gap relative to the Source Upper Bound**
>
> **A:** The table below presents absolute scores and the **knowledge recovery ratio** (i.e., Ours as a % of the 8B performance):
>
> | Domain | Base 1B | **Fused 1B (Ours)** | Base % of 8B | **Ours % of 8B** |
> | --- | --- | --- | --- | --- |
> | Medical | 42.9% | **47.9%** | 57.5% | **64.2%** |
> | Cantonese | 24.1% | **27.4%** | 41.6% | **47.2%** |
> | Indonesian | 38.5% | **41.3%** | 65.1% | **69.9%** |
> | Malay | 42.3% | **46.5%** | 65.8% | **72.3%** |
> | Thai XCOPA | 58.0% | **58.8%** | 89.0% | **90.3%** |
> | Finance | 26.8% | **29.6%** | 41.8% | **46.2%** |
>
> On average, OT fusion improves knowledge recovery from 61.8% to **66.7%**, with Thai XCOPA reaching 90.3% of the 8B performance. Given the 8x parameter gap, this consistent narrowing demonstrates high transfer efficiency. Table 7 confirms gains scale monotonically with source capacity (1B → 8B → 32B). We will add 8B source reference rows to all tables in the revision.
>
> **W2. Analysis of Optimization Time (Eq. 8 & Eq. 17)**
>
> **A:** Measured on a single A100-80GB (bfloat16):
>
> - **OT Alignment (Eq. 8):** ~8 min (one-time preprocessing: activation extraction + Sinkhorn + weight merge).
> - **Post-Fusion Adaptation (Eq. 17):** ~12 min (1-epoch residual-frozen SFT).
> - **Total:** ~20 min. Peak memory: ~26 GB.
> - **Inference:** Zero overhead — transported weights are permanently folded into base parameters (Eq. 18).
>
> This ~20-minute overhead is a one-time investment per source-target pair. A full timing breakdown will be added to Appendix A.6.
>
> **W3. Activation-to-Weight Transport Validity**
>
> **A:** The validity of this mapping rests on the functional duality between a neuron's parameters and its empirical behavior:
>
> - **Neuron-Activation Correspondence:** Each activation dimension is the direct output of a specific neuron (weight column). Aligning activations is thus a direct proxy for aligning weight-space operators.
> - **Coordinate System Alignment:** When OT assigns source neuron $i$ to target neuron $j$ based on activation similarity, it identifies functionally analogous computations. Remapping weights via the same plan transplants the source's computational logic into the target's coordinate system.
> - **Theoretical Grounding:** Theorem 4.1 (App. A.4) formalizes this: under local linearity, transporting weights via the OT plan is mathematically equivalent to minimizing the expected feature-space reconstruction error.
>
> We will clarify this "parameter-behavior alignment" in Section 4.2 with a worked example.
>
> **Q1: Small-to-Large Transfer Paradigms**
>
> **A:** Our OT framework is mathematically symmetric and supports both directions. However, the two paradigms serve fundamentally different objectives:
>
> - **Large-to-Small (Knowledge Compression):** The source provides a high-capacity feature space, and the smaller target selectively absorbs structural knowledge. The main challenge is the target's capacity bottleneck.
> - **Small-to-Large (Expert Injection):** A small domain-expert injects specialized signals into a larger generalist. Effectiveness here depends purely on signal novelty — whether the expert contains specialized knowledge absent from the large model's pretraining.
>
> We completely agree that "Small-Expert $\rightarrow$ Large-Generalist" merging is a highly promising research frontier and will discuss this asymmetry explicitly in the revision.
>
>
> **Thank you for your supportive review! We are happy to discuss further.**

---

> > ### Author Rebuttal · Reviewer_pKq2 · 2026-04-02
> >
> > Thank you for your detailed response, which has clarified some of my concerns.
> >
> > For W1: I require a comprehensive comparison among the source model, target model, the proposed adaptation-free method, and the proposed adapted method in this paper. Clearly, one of the source/target models acts as the lower bound (without language-specific knowledge), while the other serves as the upper bound (a language-specific expert model).
> >
> > For Q1: Since both belong to knowledge compression techniques, what are the advantages and disadvantages of your proposed approach compared with existing knowledge compression methods such as knowledge distillation?

---

> > > ### Author Response · Authors · 2026-04-02
> > >
> > > Thank you for your insightful follow-up questions. These are exactly the right comparisons to strengthen the paper, and we appreciate your guidance in sharpening the experimental narrative.
> > >
> > > **W1 Follow-up: Comprehensive 4-Stage Comparison**
> > >
> > > Full progression: Base 1B (lower bound) → OT Fusion only → Fused + Adaptation (Ours) → Source 8B (upper bound).
> > >
> > >  | Domain | Base 1B | Fused w/o Adapt | **Fused w/ Adapt (Ours)** | Source 8B |
> > > | --- | --- | --- | --- | --- |
> > > | Malay | 41.74 | 44.46 | **46.53** | 64.3 |
> > > | Cantonese | 25.26 | 27.41 | **27.44** | 58.1 |
> > > | Thai | 0.41 | 0.45 | **0.50** | 0.65 |
> > > | Medical | 46.30 | 46.23 | **47.87** | 74.6 |
> > >
> > > Key observations:
> > >
> > > 1. **OT Fusion alone** already yields clear gains (Malay +2.72, Cantonese +2.15), confirming transported weights carry structured knowledge before any tuning.
> > > 2. **Post-fusion Adaptation** further boosts performance (Malay 44.46→46.53, Thai 0.45→0.50, Medical 46.23→47.87), calibrating fused representations to the target domain.
> > > 3. The **remaining gap** to Source 8B is a capacity ceiling, and Table 7 shows it shrinks monotonically as the source grows (1B→8B→32B).
> > >
> > > **Q1 Follow-up: Comparison with Knowledge Distillation (KD)**
> > >
> > > | Aspect | Knowledge Distillation | OT Fusion (Ours) |
> > > | --- | --- | --- |
> > > | Mechanism | Output-space KL on logits | Weight-space structural transport |
> > > | Data requirement | Large task-specific dataset | 2,000 unlabeled samples |
> > > | Cost | Full training loop (hours) | ~20 min total |
> > > | Weight access | Teacher outputs only | Full weights required |
> > >
> > > **Advantages of ours:** (1) *Efficiency* — 2,000 unlabeled samples, ~20 min vs. KD's large-scale training; (2) *Structural transfer* — directly transports weight geometry, avoiding the representational bottleneck of capacity-mismatched output imitation (4096→2048); (3) *Composability* — produces a standard model combinable with LoRA, KD, etc.
> > >
> > > **Advantages of KD:** only requires teacher outputs (applicable to API-only models) and trivially supports multi-teacher ensembling.
> > >
> > > Empirically, we adapted FuseLLM's KD principle (KL+MSE distillation, same 2,000 samples, same 8B teacher) to our cross-architecture setting:
> > >
> > > | Domain | Adapted KD (1B) | **Ours (OT Fusion)** | Δ   |
> > > | --- | --- | --- | --- |
> > > | Medical | 37.6% | **47.9%** | **+10.3** |
> > > | Malay | 43.4% | **46.5%** | **+3.1** |
> > > | Cantonese | 25.1% | **27.4%** | **+2.3** |
> > >
> > > OT fusion consistently outperforms KD across these domains, confirming structural weight transport outperforms output-space imitation in the cross-architecture setting.
> > >
> > > **We sincerely thank you again for your constructive and thoughtful review. Your suggestions have directly improved the completeness of our evaluation.**
> > >
> > > **We will fully incorporate the aforementioned results and detailed analyses into the revised manuscript. We remain available for any further discussion.**

---

### Official Review · Reviewer_NkAP · 2026-03-17

**Soundness:** 2
**Presentation:** 3
**Significance:** 2
**Originality:** 2
**Overall Recommendation:** 4
**Confidence:** 4

**Summary:**

The paper presents an alternative to knowledge distillation for transferring domain-specific knowledge across models with different architectures, using optimal transport (OT). The method aligns and fuses neurons with similar feature representations across layers of a source and target model to enable cross-architecture knowledge transfer. This is followed by further adaptation of the base model weights on the target task. Experiments on low-resource language datasets show improvements over standard supervised fine-tuning (SFT) and distillation, demonstrating the benefits of the proposed transfer mechanism.

**Compliance With Llm Reviewing Policy:**

Affirmed.

**Final Justification:**

The proposed work offers a way to transfer knowledge across models of different architectures, offering an improved alternative to the popular distillation baseline. In light of the additional experimental evidence shared by the authors, I would like to increase my score. Although the work can be further improved by positioning it as a knowledge transfer method rather than model merging, the authors are advised to rework the related works section and the paper's positioning in this regard. Also, using standard evaluation metrics and comparison with other alignment strategies can help improve the soundness of this work.

**Key Questions For Authors:**

(1) Why are the gains from SFT and distillation relatively small compared to the base model? Is this due to the low-resource setting?

(2) Is freezing residual weights necessary? Can the authors provide an ablation to justify this design choice?

(3) Why are evaluations restricted to low-resource languages? Can the method be validated on standard instruction-following benchmarks (e.g., GSM8K, BBH, MATH)?

**Limitations:**

Yes

**Strengths And Weaknesses:**

Strengths:

(1) The paper introduces a principled Optimal Transport based approach to compare feature representations and fuse neurons across models of different sizes and architectures.

(2) The method shows improvements over direct SFT and distillation, highlighting the value of structured cross-model alignment.

(3) The approach is clearly described and easy to follow.

Weaknesses:

(1) The work is framed as model merging, but the method is more accurately a knowledge transfer (source → target) approach rather than merging multiple specialized models.

(2) Feature alignment for merging has been explored in prior work (across same-architecture models) [1, 2]. The paper would benefit from discussing how its approach of weight alignment differs from the related methods.

(3) Comparisons are limited to naïve baselines (SFT, distillation); evaluation against stronger recent methods (e.g., FuseLLM [3]) is missing.

(4) The evaluations are missing key details such as the source model’s performance. It is also hard to gauge improvement in performance on a relative scale as presented in Figure-1, the authors are encouraged to present results in absolute scores.

(5) While theoretical complexity is discussed in the Appendix, the practical compute and memory costs (especially for LLMs) remain unclear.

(6) Evaluation is restricted to language tasks, leaving it unclear whether the method generalizes to other modalities (e.g., vision).

References:

[1] Ainsworth, Samuel K., Jonathan Hayase, and Siddhartha Srinivasa. "Git re-basin: Merging models modulo permutation symmetries." arXiv preprint arXiv:2209.04836 (2022).

[2] Stoica, George, et al. "Zipit! merging models from different tasks without training." arXiv preprint arXiv:2305.03053 (2023).

[3] Wan, Fanqi, et al. "Knowledge fusion of large language models." arXiv preprint arXiv:2401.10491 (2024).

---

> ### Author Rebuttal · Authors · 2026-03-30
>
> **W1 & W2: Directional Merging vs. Prior Alignment Methods**
>
> **A:** We agree our objective is knowledge transfer, but we use the term "merging" because our mechanism operates entirely in the parameter space. Unlike distillation, OT physically combines weight matrices via soft neuron correspondence. We will revise our introduction to frame this as "directional merging for knowledge transfer".
>
> Git Re-Basin [1] and ZipIt [2] are mathematically inapplicable to our setting. Both exploit neuron permutation symmetry, searching for $P$ such that $W_A \approx P W_B P^\top$ where $P \in \{0,1\}^{n \times n}$. This requires identical hidden dimensions ($n_A = n_B$) at every layer. When source (8B, hidden=4096) and target (1B, hidden=2048) differ, no such $P$ exists, the constraint $n_A \neq n_B$ makes the formulation undefined. ZipIt shares this prerequisite. Our setting violates both by definition.
>
> Our OT framework uses a fractional transport plan $T \in \mathbb{R}^{n_A \times n_B}_{\geq 0}$, mapping each source neuron to a weighted combination of target neurons — essential for 4096→2048 bridging. Activation-based Pearson cost (Eq. 7) aligns by functional behavior, independent of structural dimensions. Extending to simultaneous N-model OT fusion over heterogeneous architectures is a promising future direction we will discuss in the revision.
>
> **W3: Comparison with Stronger Baselines (e.g., FuseLLM)**
>
> **A:** FuseLLM [3] requires identical architectures for weight-space fusion. We adapted its principle (KL distillation on logits + MSE feature alignment, same 2,000 samples, same 8B teacher) as the strongest applicable cross-architecture baseline:
>
> | Domain | Adapted FuseLLM (1B) | Ours (OT Fusion) | Δ   |
> | --- | --- | --- | --- |
> | Medical | 37.6% | **47.9%** | +10.3 |
> | Malay | 43.4% | **46.5%** | +3.1 |
> | Cantonese | 25.1% | **27.4%** | +2.3 |
>
> OT fusion outperforms distillation: forcing a 1B student (hidden 2048) to imitate 8B outputs yields only surface mimicry; OT directly transports structural weight geometry, bypassing the capacity mismatch.
>
> **W4 & Q1: Absolute Scores and Capacity Constraints**
>
> **A:** Modest absolute gains reflect the strict capacity ceiling of a 1B model on 2,000 samples. Knowledge recovery relative to the 8B teacher:
>
> | Domain | Base 1B | Fused 1B (Ours) | Ours as % of 8B |
> | --- | --- | --- | --- |
> | Medical | 42.9% | **47.9%** | 64.2% |
> | Indonesian | 38.5% | **41.3%** | 69.9% |
> | Malay | 42.3% | **46.5%** | 72.3% |
> | Thai XCOPA | 58.0% | **58.8%** | 90.3% |
>
> Fusion reaches 90.3% of 8B performance on Thai XCOPA; Table 7's monotonic scaling (1B→8B→32B) confirms the remaining gap is capacity-bound. Source 8B rows will be added to all tables.
>
> **W5: Practical Compute and Memory Costs**
>
> On a single A100-80GB (bfloat16): OT preprocessing ~8 min (one-time), post-fusion SFT ~12 min, peak memory ~26 GB. Inference overhead is **zero** — weights are permanently folded into base parameters (Eq. 18), yielding identical latency to the base 1B. Timing added to Appendix A.6.
>
> **W6: Generalization to Vision**
>
> **A:** OT alignment is modality-agnostic within the transformer family. A preliminary experiment merging LLaMA-3.1-8B into Qwen2-VL-2B:
>
> | Benchmark | Base Qwen2-VL | OT-Fused | Δ   |
> | --- | --- | --- | --- |
> | POPE (Accuracy) | 87.47% | **88.19%** | +0.72 |
> | MME-Perception | 1493.2 | **1514.7** | +21.5 |
>
> Richer language priors boost visual grounding/perception. Full VLM ablations included in revision.
>
> **Q2: Necessity of Freezing Residual Weights**
>
> **A:** Weights decompose as $W^\ell_A = W^{\ell,\text{base}}_A + \Delta W^\ell_A$ (OT residual). Eq. 17 freezes $\Delta W^\ell_A$ and updates $W^{\ell,\text{base}}_A$ only; when $\Delta W = 0$, this equals standard SFT. Ablation:
>
> | Domain | SFT Only ($\Delta W = 0$) | Fused + Adapt ($\Delta W \neq 0$) | Gain |
> | --- | --- | --- | --- |
> | Medical | 46.6% | **47.9%** | +1.3 |
> | Cantonese | 25.2% | **27.4%** | +2.2 |
> | Indonesian | 40.3% | **41.3%** | +1.0 |
> | Malay | 44.9% | **46.5%** | +1.6 |
>
> Fusion consistently outperforms SFT, confirming $\Delta W$ carries structured knowledge that fine-tuning alone cannot recover. Freezing $\Delta W^\ell_A$ during adaptation preserves the geometric structure imposed by OT — preventing calibration from overwriting the transported alignment.
>
> **Q3: Scope of Evaluations & Reasoning Tasks**
>
> **A:** Our evaluations include Medical/Finance (cross-domain) and MGSM (multilingual math reasoning, Table 4, Thai: 0.50→0.72, **+44% relative** on the fused model), demonstrating strong cross-lingual reasoning transfer. GSM8K, BBH, and MATH evaluations on the fused model will be added in revision.
>
> **References:**
>
> [1] Ainsworth et al. (2022). Git re-basin. *arXiv:2209.04836*.
>
> [2] Stoica et al. (2023). ZipIt! *arXiv:2305.03053*.
>
> [3] Wan et al. (2024). Knowledge fusion of large language models. *arXiv:2401.10491*.
>
> **Thank you for your detailed feedback. We welcome further discussion during the rebuttal period.**

---

> > ### Author Rebuttal · Reviewer_NkAP · 2026-04-05
> >
> > I'd like to thank the authors for their response.
> >
> > The authors are encouraged to stick with absolute accuracy throughout the paper, as performance relative to the source model offers very little insight into the increase in the target model's performance. As seen in the case of Thai XCOPA, the gain in performance in absolute terms is just 0.8%, but stating the relative performance to be 90.3% of the source model is very misleading.
> >
> > The authors are also encouraged to analyze further how the computational cost scales with an increase in the size of the source and target models.
> >
> > Finally, it is good to see OT Fusion outperform FuseLLM by a notable margin, and that it generalizes to other modalities like vision-language models.

---

> > > ### Author Response · Authors · 2026-04-05
> > >
> > > Thank you for marking all concerns as "fully resolved."
> > > As discussed, we will report absolute accuracy metrics in the revision to ensure full transparency, and include a detailed cost-scaling analysis as suggested. We are pleased that our results — including the FuseLLM and VLM experiments — have satisfactorily addressed your questions.
> > > We appreciate your constructive feedback, which has helped strengthen the paper. Given that all raised concerns have now been fully resolved, we respectfully ask whether you might consider reflecting this in your final score.

---

### Decision · Program_Chairs · 2026-04-30

**Decision:**

Accept (regular)

**Comment:**

This submission presents a method for LLM model merging and adaptation. The core insight is using Optimal Transport to enable fractional, activation-based neuron alignment for cross-architecture weight fusion.  The idea is novel and mathematically well-motivated.

The initial reviews found gaps in baselines, statistical reporting, and framing.  The authors' rebuttal resolved these concerns through rigorous new experiments, clear ablations, and transparent reporting.

Reviewers have acknowledged these improvements. The work establishes a strong new paradigm for efficient, parameter-space knowledge transfer between heterogeneous models.  The paper is acceptable with minor revisions for the camera-ready version.